# Light-responsive transcription factor CmOzf integrates conidiation, fruiting body development, and secondary metabolism in *Cordyceps militaris*

Jin-feng Chen,[1] Fu-ling Cheng,[1] Tong-yue Chen,[2] Yi-lan Xu,[1] Jia-mei Song,[1] Hui-min Wang,[1] Yu Zhang,[1] Xi-chuan Guo,[1] Jing Luo[1]

**ABSTRACT** *Cordyceps militaris*, an entomopathogenic fungus, produces diverse bioactive compounds. Conidial fitness and secondary metabolite levels critically influence its morphogenesis and entomopathogenicity, yet the regulatory mechanisms remain unclear. In this study, disruption of *Cmozf* severely impaired conidial development, significantly reducing conidial production. The *Cmozf*-deficient mutant (*ΔCmozf*) exhibited elevated polysaccharide and carotenoid accumulation in mycelia and accelerated fruiting body formation. Notably, *Cmwc-1*, a blue-light photoreceptor gene, was upregulated in *ΔCmozf*, whereas *Cmozf* expression was markedly suppressed in the *ΔCmwc-1* mutant. Overexpressing *Cmozf* in *ΔCmwc-1* restored conidial yield but had no effect on fruiting body development or carotenoid content. Further analysis revealed that CmOzf bound to the promoters of both *Cmwc-1* and *CmbrlA*, whereas CmWC-1 showed no binding activity to the *Cmozf* promoter. These results demonstrate that CmOzf modulates conidial development via the BrlA-AbaA-WetA central regulatory pathway and influences fruiting body development and secondary metabolite production through feedback inhibition of *Cmwc-1* expression. Our findings unveil novel signaling pathways linking conidiation, secondary metabolism, and fruiting body formation in *C. militaris*.

**IMPORTANCE** The light-responsive transcription factor CmOzf plays a pivotal role in regulating both conidial formation and secondary metabolite production in *Cordyceps militaris*, a commercially important medicinal fungus and biocontrol agent. Our study revealed that CmOzf acts as a central regulator in fungal development by (i) directly activating the central conidiation pathway via binding to the *CmbrlA* promoter, and (ii) forming a feedback loop with the blue-light photoreceptor CmWC-1 to modulate secondary metabolism. This newly identified CmOzf-CmWC-1 regulatory module represents a sophisticated light-responsive mechanism that differentially controls conidial reproduction and secondary metabolite biosynthesis. These findings provide crucial insights into how light signals are transduced to regulate fungal development and metabolism, offering valuable genetic targets for strain improvement in both biological pest control applications and pharmaceutical production.

**KEYWORDS** *Cordyceps militaris*, *Cmozf*, *Cmwc-1*, conidia, fruiting body, carotenoid

**Peer Reviewers** Ye-Eun Son, Kyungpook National University, Daegu, Republic of Korea; Zhangxun Wang, Anhui Agricultural University, Hefei, China

Address correspondence to Jin-feng Chen, chenjf@cque.edu.cn.

The authors declare no conflict of interest.

See the funding table on p. 14.

*Cordyceps militaris* is commonly known as an entomogenous fungus with medicinal and edible properties and contains various bioactive compounds, such as cordycepin, polysaccharides, and carotenoids (1). Many of these components exhibit pharmacological activities comparable to those of *Cordyceps sinensis* (2). *C. militaris* can produce both asexual conidia and sexual ascospores. On artificial culture media, conidia of *C. militaris* germinate to produce mycelia, which further develop into conidiophores that

generate additional conidia. When the conidia of entomogenous fungi adhere to the surface of host larvae, they secrete chitinases to degrade host cell walls, enabling their entry into the host body (3). The fungi can avoid the host's immune system and develop into mycelia to absorb nutrients (4). As the host is gradually depleted of nutrients, the mycelia grow out of the host cadaver and continue to differentiate into conidia in order to complete the asexual reproduction cycle. The mycelia of *C. militaris*, which exhibit a bipolar mating system, can differentiate into primordia that subsequently develop into fruiting bodies bearing asci.

It has been increasingly confirmed that light, as an information source, can significantly affect fungal growth and development (5). Different light receptors respond to various wavelengths: for example, cryptochrome, Vivid (VVD), white collar-1 (WC-1), and white collar-2 (WC-2) are responsible for blue light, while opsins detect green light, and phytochromes respond to red light and far red light (6). However, most fungi are sensitive to blue light. For instance, blue light influences hyphal knot formation, promoting fruiting body formation in *Coprinopsis cinerea* (7). *Neurospora crassa* is a model organism for studying light-regulated fungal development. The GATA-type zinc finger protein WC-1 functions as a blue-light receptor and regulates the light sensitivity and biological rhythms of *N. crassa* (8). Another light-responsive protein WC-2 interacts with WC-1 to form a functional white collar complex (WCC), which is negatively regulated by VVD (6). WCC may regulate gene expression through chromatin remodeling by interacting with the histone acetyltransferase NGF-1, while the histone methyltransferases DIM-5, SET-1, and HP-1 inhibit the expression of WCC-regulated genes (5). Photoreceptor proteins are highly conserved across species, and blue-light receptor proteins are known to be present in macrofungi. For instance, homologous proteins of WC-1 and WC-2 have been found in *Schizophyllum commune* (9, 10). In *Pleurotus ostreatus*, blue light-induced genes *Powc-1* and *Powc-2* are expressed in both primordia and fruiting bodies (11). *C. militaris* has seven light receptor proteins, such as CmWC-1, CmCRY-DASH, and CmVVD. CmWC-1 can directly bind to the promoter of the *Cmvvd* to regulate its expression, thus affecting the growth and development of *C. militaris* (12, 13). Recently, a new zinc-finger domain-containing protein (CmWC-3) was shown to interact with CmWC-1 and CmVVD (14).

Fungi deploy aerial hyphae into the atmosphere to perceive the external environment. Stimulated by light and oxygen, they transition from nutritional growth to reproductive growth, promoting asexual development and fruiting body formation (15). In *Beauveria bassiana*, there exists a conserved regulatory pathway for conidiation, the BrlA-AbaA-WetA pathway (16). This pathway is regulated by numerous upstream factors, such as FluG, FlbA-E (17). Blue light can upregulate the conidiophore development gene *brlA* in *A. nidulans* (18). Further research has demonstrated that the photoreceptive complex consisting of FphA, LreA (WC-1), LreB (WC-2), and VeA can regulate the transcription levels of *flbA*, *flbB*, and *flbC*, thereby activating *brlA* expression and triggering conidiation (19, 20). In *C. militaris*, light signals influence fungal development by regulating the expression of specific genes, such as *CmflbB* and *CmflbC* (21). In addition to inducing morphological changes, blue light stimulates pigment biosynthesis in *Terana caerulea* and *Monascus* spp., while also enhancing exopolysaccharide production in *Aspergillus niger* (22–24). Inactivation of WC-1 in *C. militaris* can reduce carotenoid and cordycepin contents (25). However, the downstream signaling components of the photoresponse in *C. militaris* remain elusive. In the related fungus *B. bassiana*, BbSmr1 regulates oosporein synthesis and conidial development (16). We identified its structural homolog in *C. militaris*, CmOzf (OZF: orange pigmentation-related zinc finger protein), which shares 84.6% sequence identity with BbSmr1. Notably, CmOzf expression is light-responsive and correlates with carotenoid biosynthesis and conidial development. Given these parallels, we hypothesized that CmOzf integrates light signals to coordinate key developmental processes, including conidiation, fruiting body morphogenesis, and secondary metabolite production.

## RESULTS

### *Cmozf* is highly expressed under light exposure

The coding sequence of the CmOzf gene spans 1,425 bp and encodes a 474-amino acid protein belonging to the $C_2H_2$ zinc finger protein family. Sequence analysis revealed that CmOzf contained a nuclear localization signal (NLS; KKKW, residues 239–242) and four $C_2H_2$-type zinc finger motifs ($CX_{2-4}CX_3FX_5LX_2HX_{3-5}H$) located at positions 261–283, 275–300, 291–311, and 319–338. The predicted molecular weight of CmOzf is 53.4 kDa, with a theoretical isoelectric point (pI) of 9.28. Light signal significantly affected conidial development in *C. militaris*. Conidial yield of the wild type (WT) was 5.7 times higher under light conditions than under dark conditions ($P < 0.001$), and the expression level of the *Cmozf* gene was significantly upregulated by 3.5 times ($P < 0.001$) (Fig. 1).

### Reduced conidial and blastospore yield, increased pigment, and affected fruiting body development due to *Cmozf* deletion

To investigate the relationship between the *Cmozf* gene and light signaling pathways, one segment of the *Cmozf* gene was partially replaced with a *bar* cluster to generate mutant strains. *Cmozf* deletion mutant isolates (*ΔCmozf*) were confirmed by PCR and reverse transcription-polymerase chain reaction (RT-PCR) analyses (Fig. S1). After 20 days of incubation on potato dextrose agar (PDA) medium, *ΔCmozf* mutants exhibited enhanced pigmentation, with colonies displaying a deeper orange hue compared to the WT (Fig. 2A). The WT conidial yield reached $8.3 \times 10^7$ conidia/colony. In contrast, the yields for *ΔCmozf*-7 and *ΔCmozf*-11 were only $0.11 \times 10^7$ and $0.15 \times 10^7$ conidia/colony, respectively ($P < 0.001$; Fig. 2B). Scanning electron microscopy results showed that the WT had a high number of conidia. Contrarily, the mutants primarily contained mycelia, with very few conidia (Fig. 2C). When inoculated in a nutrient-rich 1/4 SDB liquid medium for 4 days, the WT strain exhibited a turbid suspension. In contrast, the *ΔCmozf* mutant strains presented a granular mycelial ball morphology (Fig. S2A). Light microscopic observations revealed that the WT strain had both blastospore and mycelia. The mutant strains showed impaired blastospore development, primarily existing in the form of mycelium (Fig. S2B). Upon inoculation of the *ΔCmozf* mutant strains on the rice medium, the fruiting bodies of *ΔCmozf*-7 and *ΔCmozf*-11 exhibited elongation, with an average height of $3.2 \pm 0.5$ cm and $3.5 \pm 0.7$ cm, respectively (Fig. 3). Their heights were significantly greater than the average height of the WT fruiting bodies ($2.1 \pm 0.3$ cm) ($P < 0.05$).

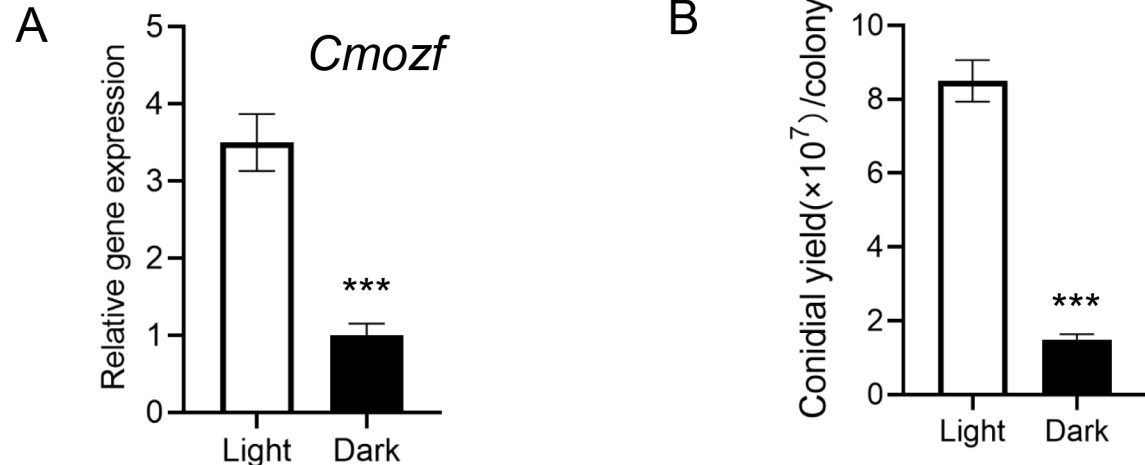

**FIG 1** Effects of light on *Cmozf* gene expression and conidial production. (A) *Cmozf* was highly expressed under light exposure. (B) Conidial yields were measured. The wild-type (WT) strain was cultured either in constant darkness for 7 days (control) or in darkness for 3 days, followed by 4 days of light exposure (treatment). ***, $P < 0.001$.

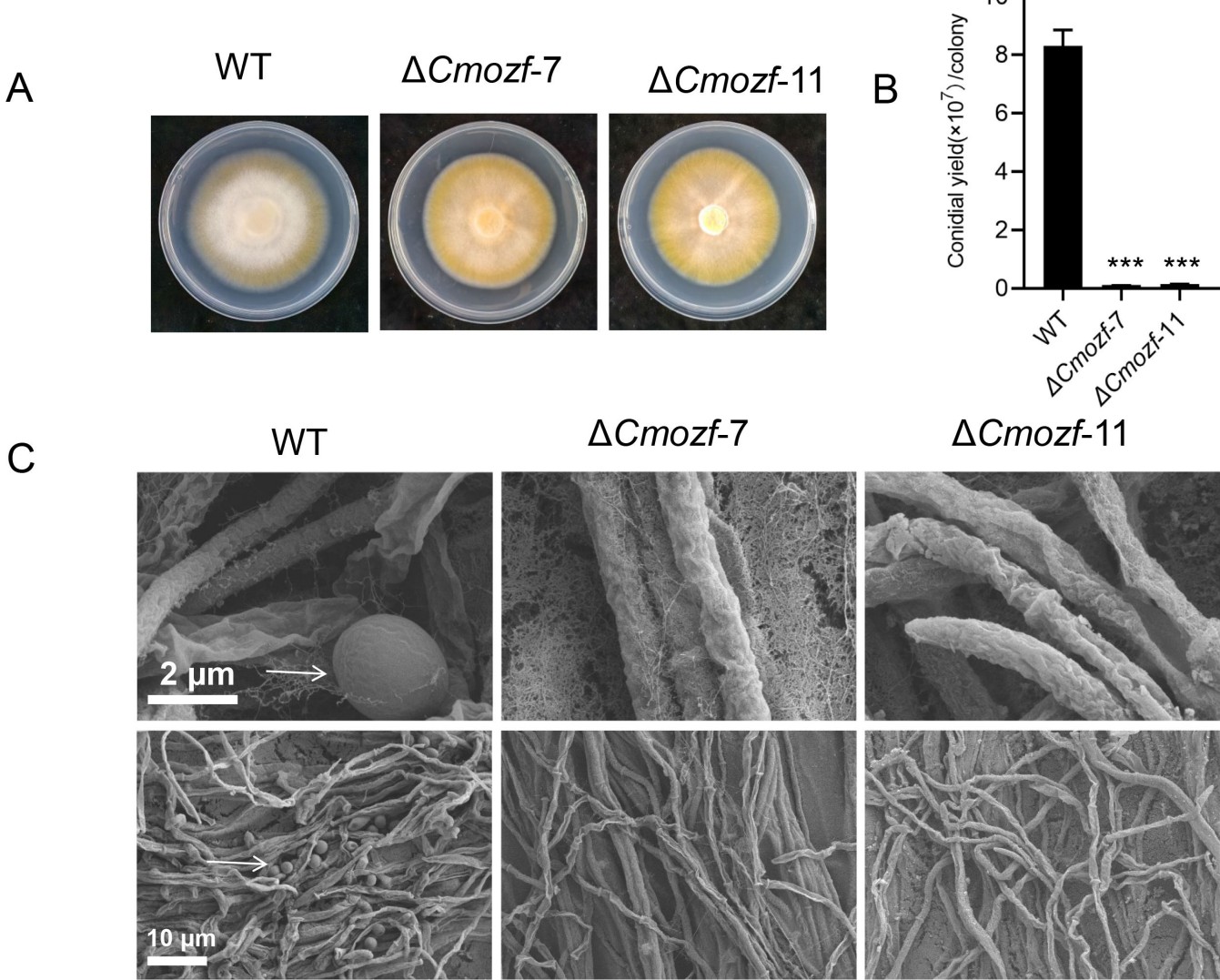

**FIG 2** Colony growth of *Cmozf* deletion strain on PDA medium for 20 days. (A) The mycelial blocks were inoculated onto PDA medium, incubated in the dark for 3 days, and then exposed to light for 17 days. (B) Analysis of conidia yield of the colonies. (C) Scanning electron micrographs of the colonies. Conidia and mycelium were observed under 20,000× (scale bars, 2 µm), and 2,500× (scale bars, 10 µm), respectively. The white arrow marks the position of formed conidia. ***, $P < 0.001$.

## Increased carotenoid and polysaccharide contents in Δ*Cmozf* mutants

After inoculation of the Δ*Cmozf* mutant strains onto the PDB medium and shaking for 4 days, mycelial pellets were obtained and then statically cultured under light exposure for 10 days. The Δ*Cmozf* mutants exhibited a deepened color and a carotenoid content approximately 2.3-fold higher than the WT ($P < 0.01$; Fig. 4A through C). The expression levels of three genes that affected the biosynthesis of carotenoids were analyzed. The mRNA level of CCM_00310 (hydroxymethylglutaryl-CoA synthase) and CCM_09155 (fatty aldehyde dehydrogenase) was upregulated by approximately onefold ($P < 0.01$), and the expression of CCM_04199 (uncharacterized gene) increased by threefold in the mutant strains ($P < 0.001$; Fig. 4D through F). Furthermore, *Cmozf* knockout increased the polysaccharide content in the mutants to 2.5-fold that of the WT ($P < 0.01$) (Fig. 5A). The mutant strains exhibited significant upregulation of two key polysaccharide biosynthesis enzymes: galactokinase (CCM_00637, 2.0-fold; $P < 0.01$) and galactose-1-phosphate uridylyltransferase (CCM_09317, 1.5-fold; $P < 0.05$) (Fig. 5B and C).

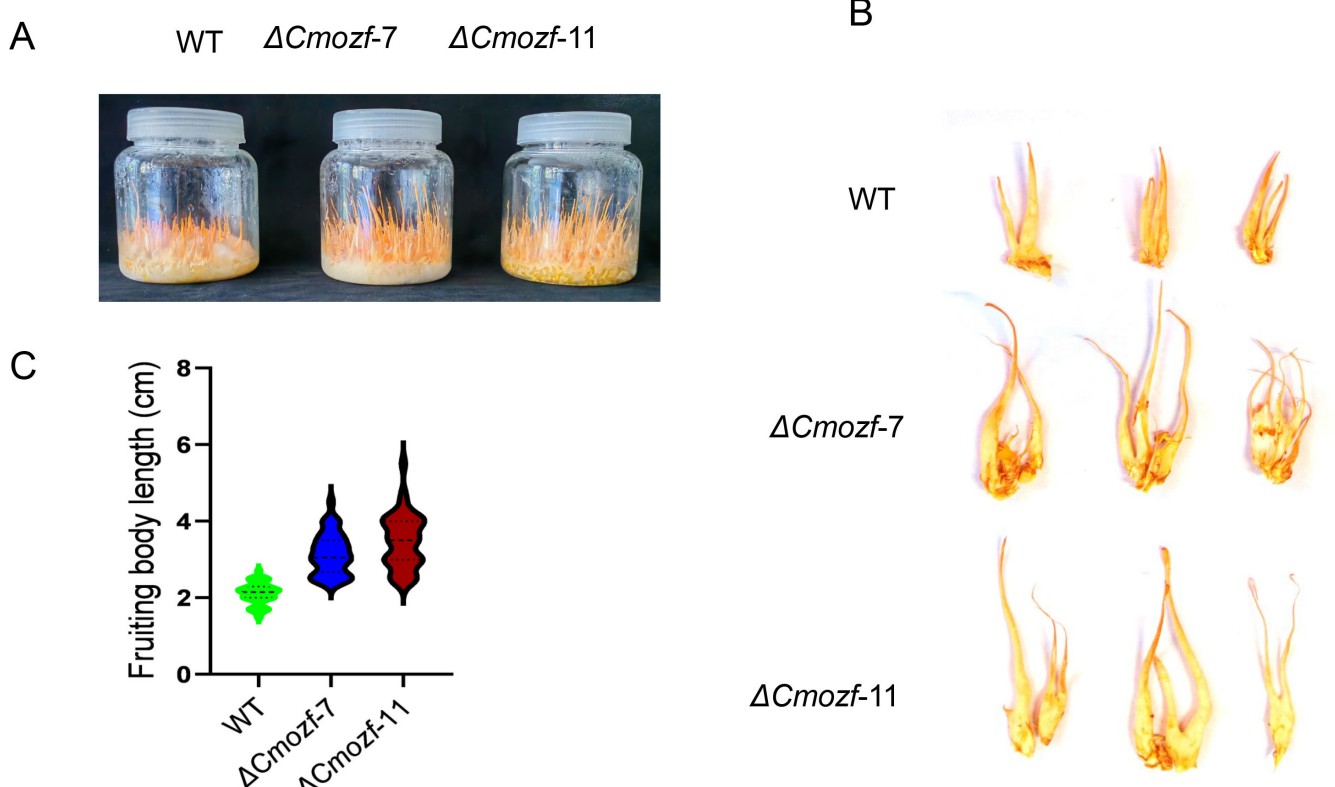

**FIG 3** Growth of fruiting bodies of *ΔCmozf* cultured on rice medium for 30 days. (A) Mycelial blocks were inoculated into plastic bottles containing rice medium, dark incubated for 7 days, and then exposed to light for 23 days before photographing. (B) Randomly selected samples were taken from the cultured fungi for photography. (C) Statistical analysis of the length of the fruiting bodies.

## Transcriptome analysis of *ΔCmozf* and *ΔCmwc-1*

CmWC-1, the most important photoreceptor, plays crucial roles in light signaling pathways. To investigate the relationship between *Cmozf* and light signaling, we generated a knockout of the blue light sensor by disrupting *Cmwc-1* (Fig. S3). Transcriptome sequencing was performed on the *ΔCmozf* and *ΔCmwc-1* mutants cultured in the dark on PDA medium for 3 days, followed by 4 days of light exposure, revealing significant changes in gene expression compared to the WT. *ΔCmozf* had a total of 1,543 differentially expressed genes (DEGs), with 818 upregulated genes and 725 downregulated genes. *ΔCmwc-1* had 685 DEGs, including 432 upregulated genes and 253 downregulated genes (Fig. 6A). Notably, 9.46% of upregulated and 6.77% of downregulated DEGs showed overlapping expression patterns between the two transcriptome data sets (Fig. 6B). Functional enrichment analysis of overlapping DEGs revealed distinct functional patterns: upregulated genes were primarily associated with membrane-related processes, including transmembrane transport (particularly inorganic molecule transport) and membrane structural organization, while downregulated genes showed enrichment in membrane integrity maintenance, transporter activity, and redox processes involving monooxygenase and oxygen-dependent oxidoreductase functions (Fig. 6C and D).

Importantly, both DEGs included key genes, such as *Cmozf*, *Cmwc-1*, *Cmvvd,* and conidial development-related genes *CmbrlA*, *CmabaA*, and *CmwetA*. To further verify these transcriptomic results, quantitative reverse transcriptase PCR (qRT-PCR) was employed to quantify the expression levels of *Cmwc-1* in the *ΔCmozf* mutant and *Cmozf* in the *ΔCmwc-1* mutant. The results showed that a 2.1-fold downregulation of *Cmozf* expression in the *ΔCmwc-1* mutant ($P < 0.001$; Fig. 7A). In contrast, the expression levels of *Cmwc-1* and *CmvvD* were upregulated by 2.75 times and 2.6 times, respectively, in the *ΔCmozf* mutant ($P < 0.001$; Fig. 7B and C).

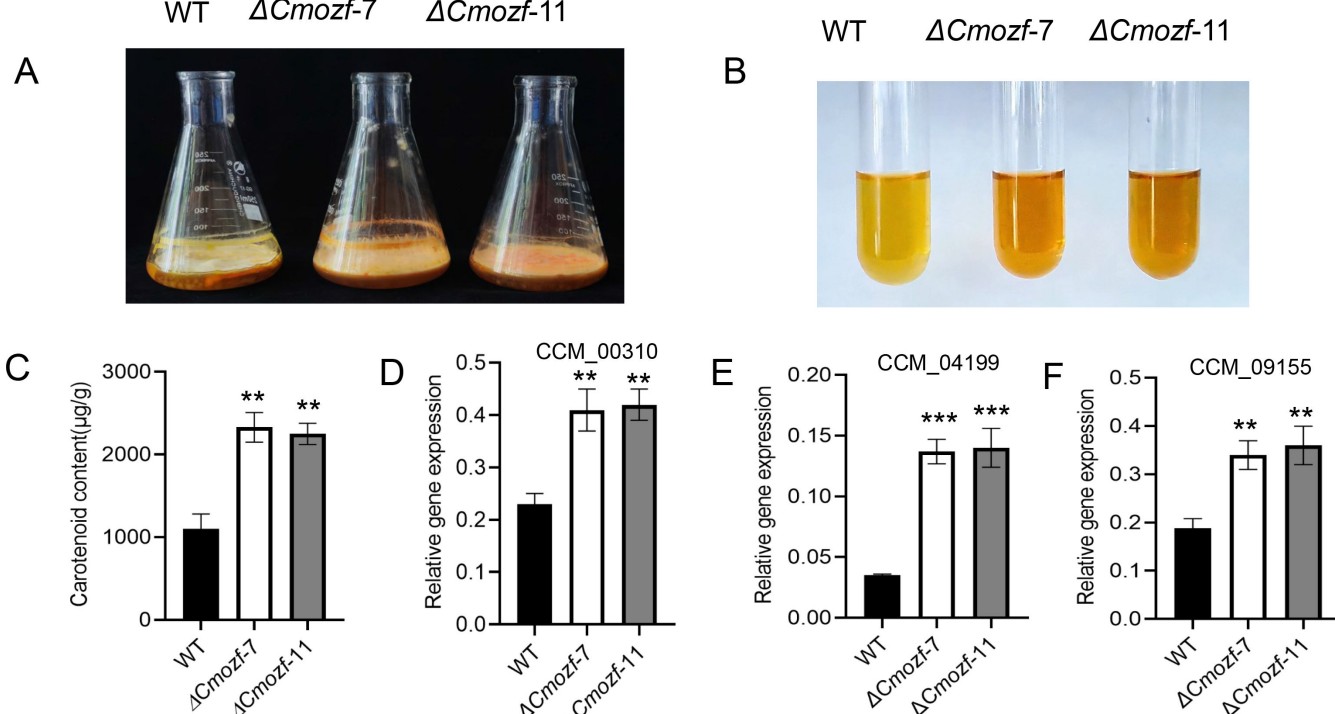

**FIG 4** Analysis of carotenoid content in the *ΔCmozf* mutant. (A) Changes in the color of the strain after 5 days of shaking culture in PDB, followed by 10 days of static culture. (B) Color intensity of the extracted samples, which is used as an indication of carotenoid concentrations. Analysis of carotenoid contents (C) and expressions of related genes (D–F). **, $P < 0.01$; ***, $P < 0.001$.

## Increased conidial production due to overexpression of *Cmozf* in *ΔCmwc-1*

To further investigate the relationship between *Cmozf* and *Cmwc-1*, the *Cmozf* gene was overexpressed in WT and the *ΔCmwc-1* mutant backgrounds, resulting in the OE*Cmozf* and *ΔCmwc-1*/OE*Cmozf* strains, respectively. Among these, Strain #1 (*ΔCmwc-1*/OE*Cmozf*) and Strain #6 (OE*Cmozf*), which showed high *Cmozf* expression, were selected for subsequent experiments (Fig. S4). After 15 days of incubation on the PDA medium. Compared to the WT strain, the *OECmozf* strain produced significantly more conidia ($P <$

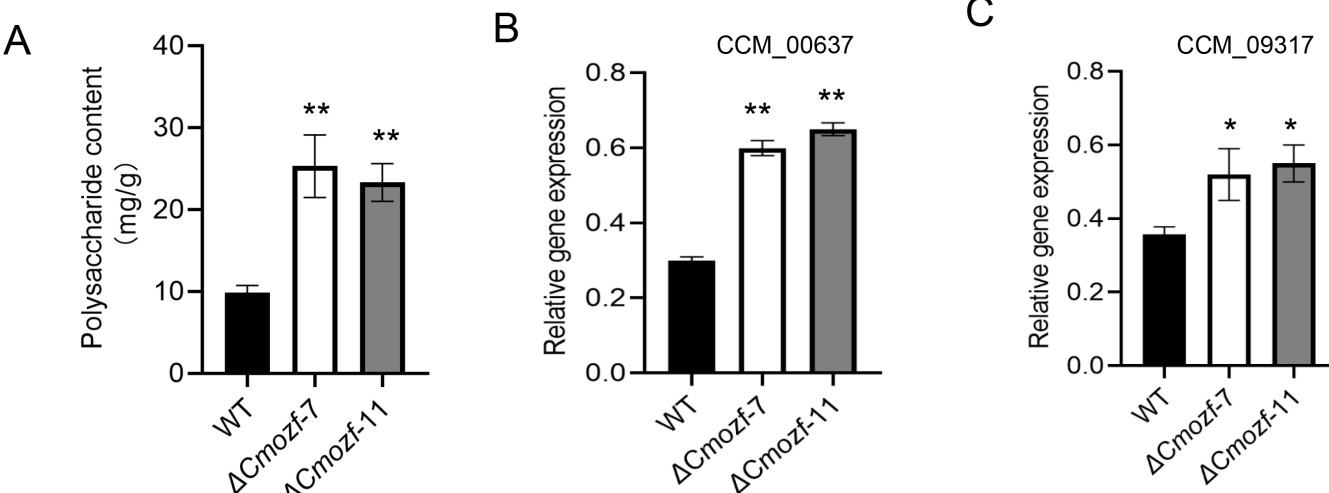

**FIG 5** Analysis of polysaccharide production in the *ΔCmozf* mutant. (A) Detection of polysaccharide content. (B and C) Expression analysis of genes related to polysaccharide synthesis. *, $P < 0.05$; **, $P < 0.01$.

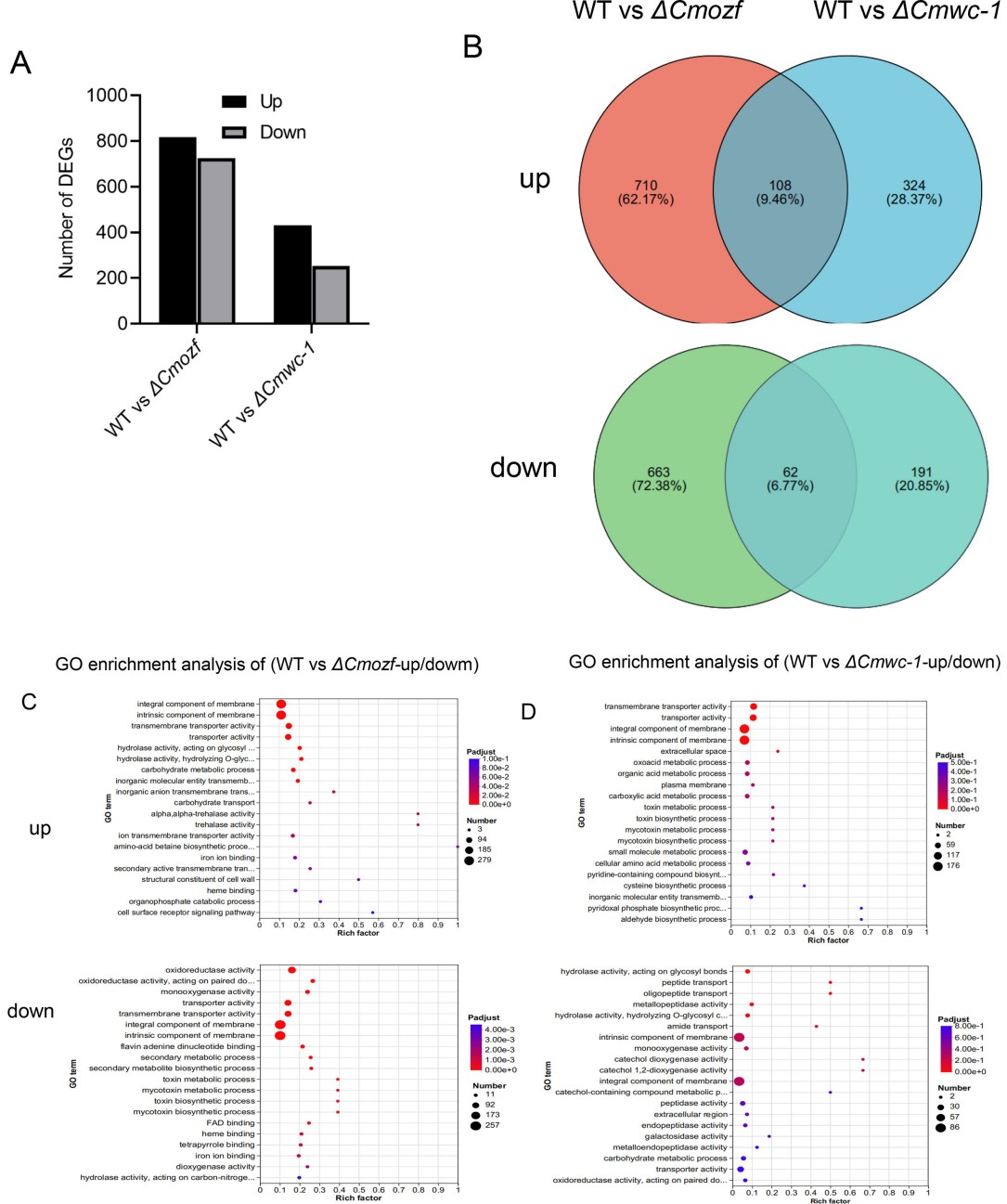

**FIG 6** Comparative transcriptomic analysis of *Cmozf* and *Cmwc-1* deletions in *C. militaris*. (A) DEGs of *ΔCmozf* and *ΔCmwc-1* compared with WT. (B) Venn diagram showing up- and down-regulated genes across mutant strains. The number of genes in the overlapping area represented the genes expressed in both mutants. (C and D) Gene ontology (GO) classification of DEGs. X-axis represented the enrichment factor. Y-axis represented the pathway name. Coloring indicated the q-value (high: blue; low: red), and the lower q-value indicates the more significant enrichment. The point size indicates the number of DEGs (more: big; less: small).

0.05) but accumulated fewer carotenoids in the mycelium ($P < 0.05$) (Fig. 8A through C). *ΔCmwc-1*/OE*Cmozf* strain partially reversed the fluffy phenotype of the *ΔCmwc-1* mutant (Fig. 8A). Accordingly, the conidial yield increased from $0.36 \times 10^7$ conidia/colony to $3.6 \times 10^7$ conidia/colony, reaching about half of the WT conidial production ($P < 0.001$; Fig. 8B). However, carotenoid levels and undifferentiated primordium in the *ΔCmwc-1*/OE*Cmozf* strain did not significantly change compared to the *ΔCmwc-1* strain ($P > 0.05$; Fig. 8C; Fig. S5). Expression levels of *CmbrlA* and *CmabaA* were significantly elevated in both the

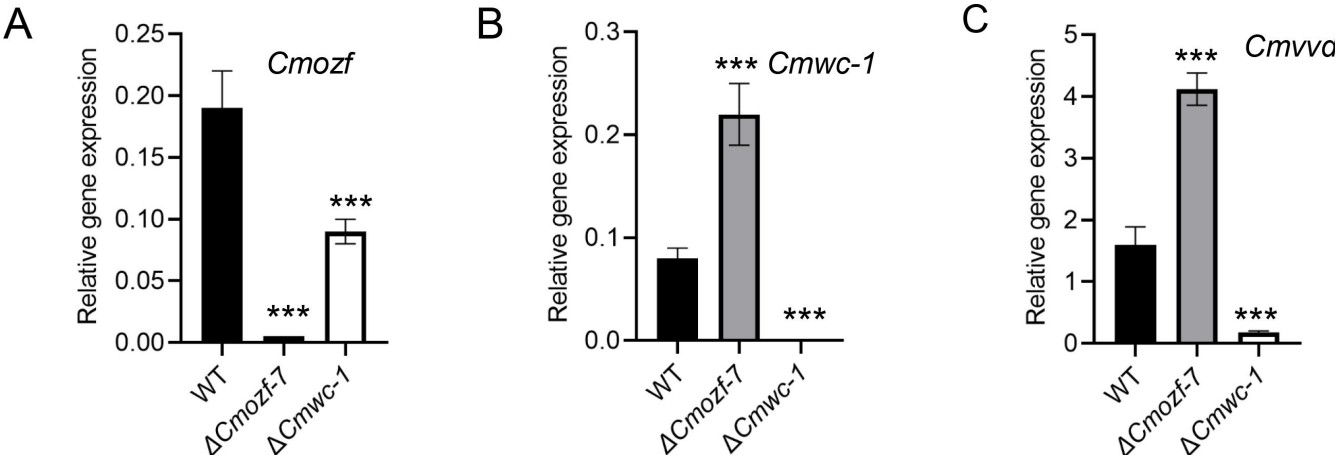

**FIG 7** Expression analysis of genes in *ΔCmozf* or *ΔCmwc-1* strains. Total RNA was extracted from fungal strains cultured on PDA medium in the dark for 3 days, followed by exposure to light for 4 days, and then used for real-time PCR analysis. Relative expression levels of *Cmozf* (A), *Cmwc-1* (B), and *Cmvv*d (C) were measured. ***, $P < 0.001$.

OE*Cmozf* strain relative to WT ($P < 0.05$) and the *ΔCmwc-1*/OE*Cmozf* strain relative to *ΔCmwc-1* ($P < 0.001$; Fig. 8D and E).

## CmOzf negatively regulates *Cmwc-1* expression

To further confirm the relationship between *Cmozf* and *Cmwc-1*, yeast one-hybrid assays were conducted. The coding sequences of *Cmozf* and *Cmwc-1* were cloned into the pB42AD vector. The DNA sequences at about 1,000 bp upstream of the start codons ATG of *Cmozf*, *Cmwc-1*, and *CmbrlA* were separately cloned into the placZi vector as promoter sequences. The results showed that CmOzf bound to the promoters of *Cmwc-1* and *CmbrlA* (Fig. 9A; Fig. S6A). However, the promoter of *Cmozf* could not be bound by

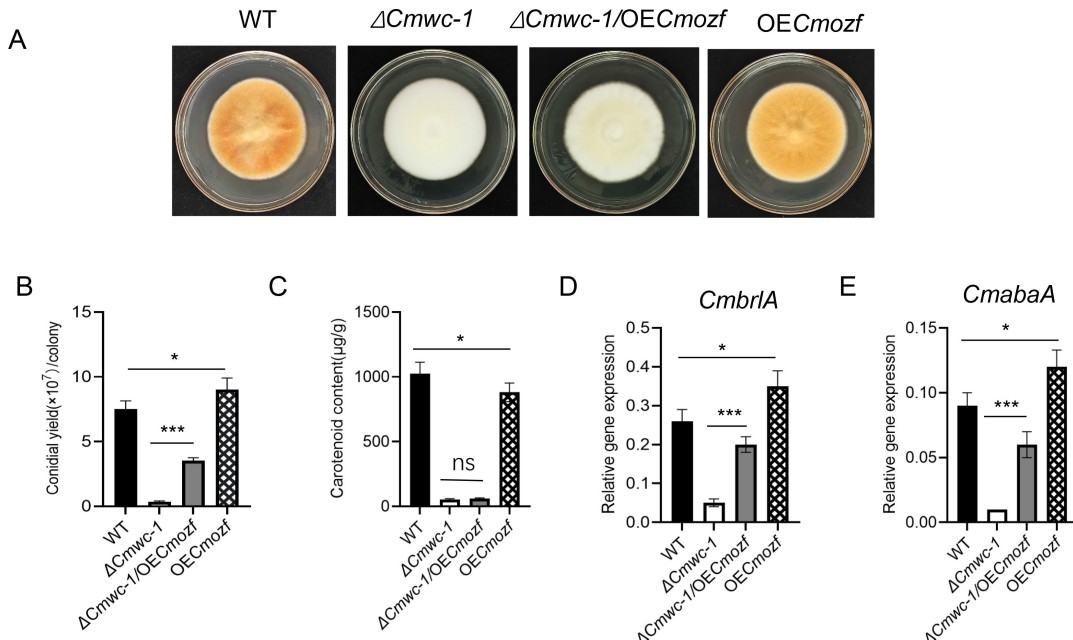

**FIG 8** Overexpression of *Cmozf* partially rescues conidiation defects in *ΔCmwc-1* strain. (A) Conidial suspensions of WT, the *Cmwc-1* deletion strain (*ΔCmwc-1*), *Cmozf*-overexpressing strains (*ΔCmwc-1*/OE*Cmozf* and OE*Cmozf*) were inoculated on the PDA medium, respectively. After culturing for 3 days in the dark, followed by exposure to light for 12 days, the phenotypes were observed. Conidial yields (B) and carotenoid contents (C) were measured. Expressions *of CmbrlA* (D) and *CmabaA* (E) were analyzed in strains cultured in the dark for 3 days, followed by 4 days of light. ns, not significant; *, $P < 0.05$; **, $P < 0.01$; ***, $P < 0.001$.

CmWC-1. Furthermore, the promoter of *Cmwc-1* was divided into three segments: P1 (−432 to −742 bp), P2 (−153 to −441 bp), and P3 (−1 to −164 bp). It was also found that only segments P2 and P3 could be bound by CmOzf (Fig. 9B). The interaction between the transcription factor and the target promoter was shown in Fig. S6B. To further elucidate the relationship between CmWC-1, CmOzf, and the central regulatory pathway of conidial development, we analyzed the gene expression levels of *Cmwc-1*, *Cmozf*, and three key regulator genes in the conidiation pathway over a period of 4 to 10 days. The results revealed that the expression levels of *CmbrlA*, *CmabaA*, and *CmwetA* peaked at day 7 before subsequently declining in the WT strain. In the Δ*Cmozf* and Δ*Cmwc-1* strains, compared to WT, the expression levels of *CmbrlA* and *CmabaA* were significantly reduced at all time points (Fig. 10A through C). In the Δ*Cmwc-1* mutant, the expression of *Cmozf* was significantly decreased at all time points relative to WT (Fig. S7A). Interestingly, deletion of *Cmozf* led to a significant upregulation of *Cmwc-1* expression at days 7 and 10 (Fig. S7B).

## DISCUSSION

### CmOzf affects developmental differentiation in *C. militaris*

CmOzf shared 86.0% amino acid homology with the BbSmr1 of *B. bassiana*, both containing four zinc finger protein domains. Deletion of *Bbsmr1* impaired conidial development and reduced conidial yield (16). Similarly, the Δ*Cmozf* strain showed significant impairment in conidial development with negligible conidia. The homologous proteins of COS1 from *Magnaporthe oryzae* and CgAzf1 from *Colletotrichum gloeosporioide*s have also been associated with conidial development (26, 27). This suggests that the regulation of conidial development pathways is conserved among these filamentous fungi.

### CmOzf responds to light signaling pathways

Light is a critical environmental factor for the development of many fungi (28). Light exposure significantly increased conidial production in *C. militaris* and the expression level of *Cmozf*. Thus, the deletion of *Cmozf* significantly reduced conidial yield. BrlA and AbaA are recognized as key control points for conidiophore development and conidial maturation in many filamentous fungi, respectively (16, 29). The expression levels of both *CmbrlA* and *CmabaA* were significantly downregulated in the Δ*Cmozf* strain. The photoreceptor protein WC-1 can regulate blue light genes (30). It has been found that sporulation decreases in the WC-1 ortholog LreA mutant strains in *Alternaria alternata* (31). CmWC-1 regulates the fruiting body development and secondary metabolism in *C. militaris* (25). Knockout of the *Cmwc-1* also significantly reduced conidial production. Correspondingly, the expression levels of *Cmozf*, *CmbrlA*, *CmabaA*, and *CmwetA* all decreased in the Δ*Cmwc-1* strain. Yeast one-hybrid assays confirmed that CmOzf could bind to the *CmbrlA* promoter. CmOzf is a key regulator of conidial development, mirroring the role of its homolog BbSmr1 in hyphal differentiation and

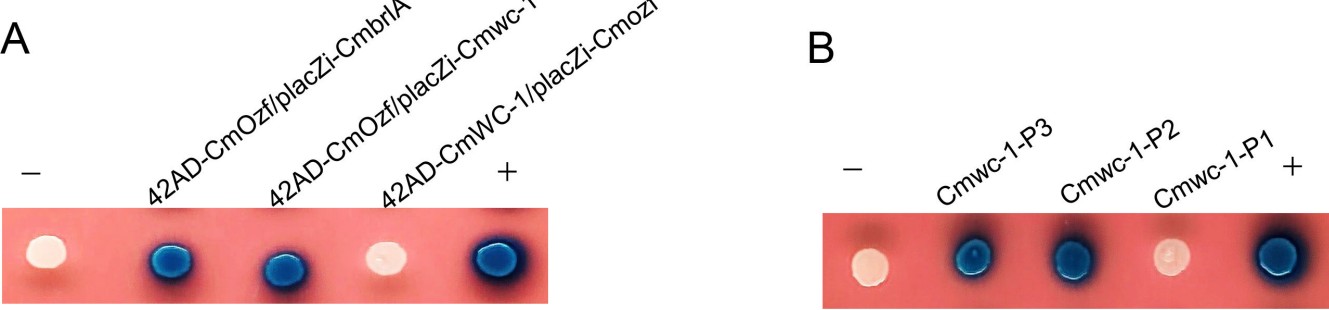

**FIG 9** Yeast one-hybrid analysis of transcription factor and promoter interactions. (A) Interaction of CmOzf with the promoters of *CmbrlA* and *Cmwc-1*, respectively. (B) CmOzf bound to different fragments of the *Cmwc-1* promoter.

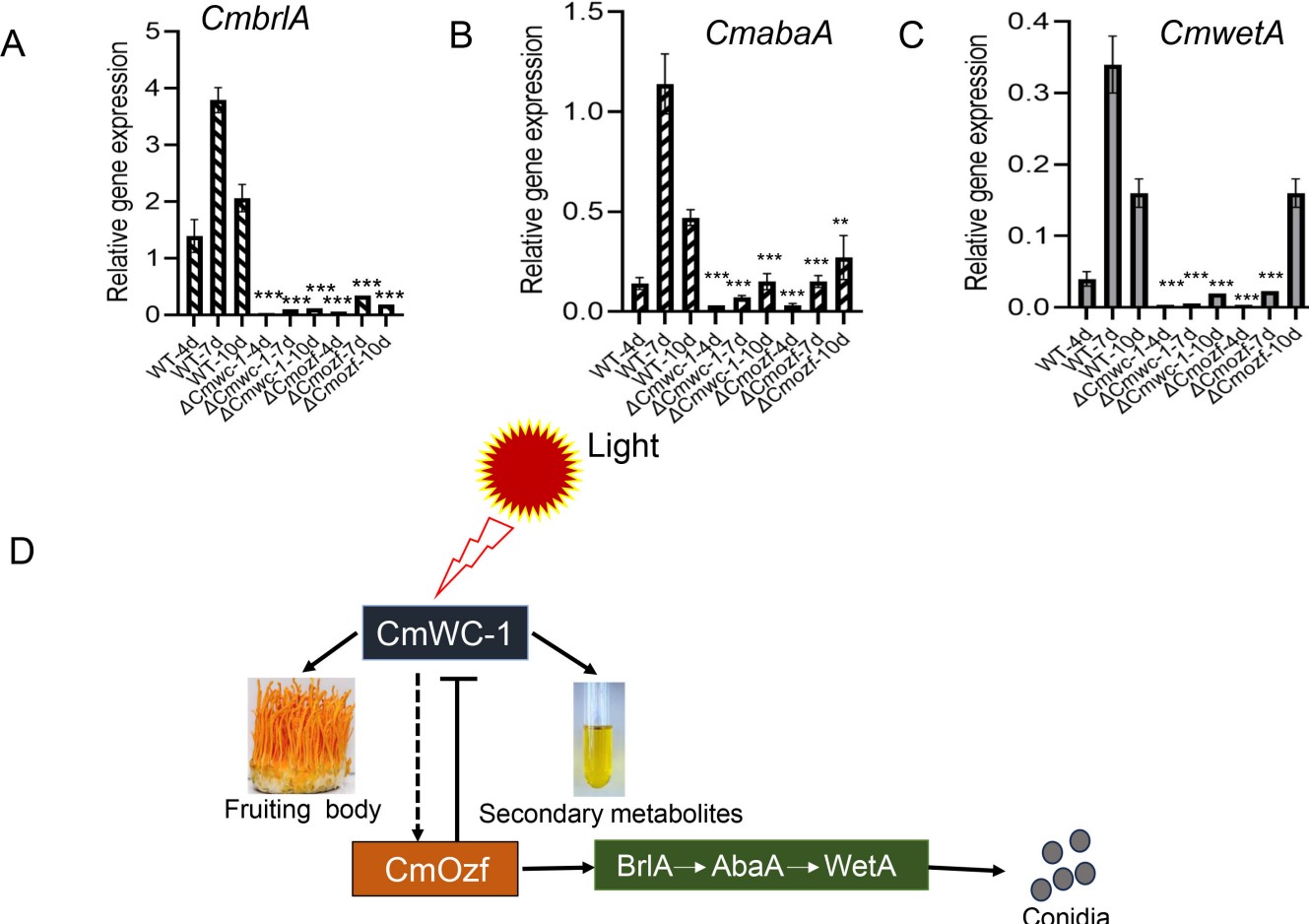

**FIG 10** Regulatory roles of CmWC-1 and CmOzf in conidiation, fruiting body, and secondary metabolism. (A–C) mRNA levels of *CmbrlA*, *CmabaA*, and *CmwetA* in *ΔCmwc-1* and *ΔCmozf* strains were analyzed. Samples were collected after 3 days of dark incubation, followed by 1, 4, and 7 days of continuous light on PDA medium. (D) Proposed model of CmOzf-mediated regulation of development and secondary metabolism. CmWC-1 regulates conidiophore formation through the CmWC-1–CmOzf–CmBrlA pathway. CmOzf exerts feedback inhibition on CmWC-1 to regulate secondary metabolite production, fruiting body development. **: $P < 0.01$; ***: $P < 0.001$.

conidial maturation (16, 32). However, it remains unclear how CmWC-1 interacts with other proteins to regulate the expression of *Cmozf*.

### *CmOzf* regulates secondary metabolite synthesis and fruiting body development

The orange pigments in *C. militaris* are generally considered to be carotenoids, and carotenoid biosynthesis is light-induced (33). A few genes might be involved in the carotenoid biosynthesis in *C. militaris*, such as *Cmtns*, CCM_00310, and CCM_09155 (34). Deletion of *Cmozf* also upregulated the expression of these genes, resulting in increased carotenoid content in the *ΔCmozf* strain. The high levels of carotenoids and polysaccharide content in the *ΔCmozf* strain have significant implications for protecting the fungus against host defenses, but the specific mechanisms involved still need to be further determined. The fruiting body development of the *ΔCmozf* was accelerated, growing longer and slenderer compared with the WT. However, targeted deletion of the *Cmwc-1* gene induced mycelia albinism and primordial formation failure under light. Notably, even under *Cmozf* overexpression in the *ΔCmwc-1* background, carotenoid production and fruiting body development were not altered in the *ΔCmwc-1* mutant. This suggests divergent regulatory pathways for conidial development versus secondary

metabolism and fruiting body formation. Unfortunately, due to the low conidial yield of the ΔCmozf mutant and the potential lethality of double knockout mutants, we have not yet obtained the ΔCmozf/ΔCmwc-1 double knockout mutant. Interestingly, yeast one-hybrid assays confirmed that CmOzf could bind to the Cmwc-1 promoter to inhibit its gene expression. The feedback inhibition phenomenon was also common in fungi (35). Cmvvd is the direct target of CmWC-1, deletion of the Cmvvd gene leads to abnormal fruiting body development and a significant increase in carotenoid production (13). The ΔCmozf and ΔCmvvd mutants displayed similar phenotypes. However, the expression level of Cmvvd was upregulated in the ΔCmozf mutant. This suggests that there may be a feedback inhibition loop between Cmozf and Cmwc-1 regarding conidial development, carotenoid production, and fruiting body development.

In summary, our experiments confirmed that Cmwc-1 responded to light signals via Cmozf to activate the central pathway for conidial development. Cmozf also exerted direct feedback inhibition on Cmwc-1 expression to regulate secondary metabolism and fruiting body development in C. militaris (Fig. 10D). These results can help to further clarify the molecular mechanisms underlying the growth and metabolism of C. militaris.

## MATERIALS AND METHODS

### Microbial strains, media, and cultivation conditions

A commercial strain of C. militaris with a round head fruiting body (Zhou Yulin Edible Fungi Research Institute, Wuhan, China) was used in this study. The strains were inoculated in PDA or 1/4 SDB medium (1% glucose, 0.25% peptone, and 0.5% yeast extract) at 25°C. The strains were also inoculated onto the rice medium (30 g rice, 0.9 g glucose, 0.135 g $KH_2PO_4$, 0.0675 g $MgSO_4 \cdot 7H_2O$, and 45 mL $ddH_2O$) to produce fruiting bodies. Escherichia coli DH5α was used to construct plasmids. The Agrobacterium tumefaciens strain AGL-1 was cultivated in the YEB medium (1% glucose, 0.1% yeast extract, 0.5% sucrose, 0.05% $MgSO_4 \cdot 7H_2O$, 1.5% agar, wt/vol).

### Construction of targeted gene knockout strains

Using C. militaris genomic DNA as a template, the left flanking sequence (~1,600 bp) and right flanking sequence (~1,500 bp) of the Cmozf gene (locus_tag, CCM_03777) were amplified using the primers Cmozf-left-F/R and Cmozf-right-F/R (Table S1), respectively. The left and right flanking sequences were cloned into the EcoRI and XbaI sites of the pK2-PtrpC-bar-Trptc vector using a homologous recombination enzyme (Vazyme Biotech Co., Ltd., China), respectively, in order to produce the targeted gene-deletion vector pK2-bar-Cmozf. The homologous replacement strategy was employed to disrupt the Cmozf gene by replacing the bar cassette (PtrpC+bar+TtrpC, ~1.6 kb) at the position of the coding sequence (341 bp) of Cmozf and its upstream 189 bp promoter sequence. The left (~1,500 bp) and right (~2,000 bp) flanking sequences of the Cmwc-1 gene were amplified using the primers Cmwc-1-left-F/R and Cmwc-1-right-F/R (Table S1), respectively. The pK2-bar-Cmwc-1 vector was transformed into C. militaris to delete the Cmwc-1 by replacing the internal ~1,400 bp segment of the Cmwc-1 gene with the bar cassette. The ΔCmwc-1 mutant was verified by PCR amplification with the intronic primer Cmwc-1-F/R (Table S1) and RT-PCR.

### Construction of constitutive strains

The cDNA sequences of Cmozf under the control of the promoter of the glyceralde-hyde-3-phosphate dehydrogenase gene (Bbgpd) (36) were recombined through PCRs with the primer pair pb3-F/OECmOzf-R (Table S1). The fused DNA fragment was inserted into the EcoRI site of pK2-PtrpC-sur-TtrpC to generate the pK2-sur-OECmozf vector.

## Transformation of *C. militaris*

The constructed vectors were transformed into *C. militaris* using the *Agrobacterium tumefaciens*-mediated transformation method as previously described (37, 38), with some modifications. The strain was resuspended in sterile 0.05% Tween-80 and filtered through double-layered lens cleaning papers to obtain a conidial suspension ($5 \times 10^5$ conidia/mL). This suspension was mixed with an equal volume of pre-induced AGL-1 ($OD_{600}$ = 0.15) in the IM medium (g/L) [$K_2HPO_4$ 2.28, $KH_2PO_4$ 1.36, $FeSO_4\cdot7H_2O$ 0.0025, NaCl 0.15, 2-(N-morpholino) ethanesulfonic acid 8.53, $MgSO_4\cdot7H_2O$ 0.49, $(NH_4)_2SO_4$ 0.53, $CaCl_2$ 0.07, glycerol 5.0, glucose 2, acetosyringone 0.038] for 6 h. This mixture was then uniformly spread on the IM medium supplemented with a microporous filter membrane (pore size = 0.8 μm) and co-cultured for 72 h. Subsequently, the filter membrane was transferred to the M-100 medium supplemented with 250 μg/mL cefotaxime and 0.1% phosphinothricin (PPT) (Chinese Academy of Agricultural Sciences, China) (g/100 L) (glucose 1.0, $KNO_3$ 0.3, M-100 salt solution [$H_3BO_3$ 0.06‰, $MnCl_2\cdot4H_2O$ 0.14‰, $ZnCl_2$ 0.4‰, $Na_2MoO_4\cdot2H_2O$ 0.04‰, $FeCl_3\cdot6H_2O$ 0.10‰, $CuSO_4\cdot5H_2O$ 0.4‰] 6.25 mL, agar powder 1.5 g) for culturing. Then, single resistant colonies were isolated on a 48-well plate containing the M-100 medium supplemented with cefotaxime and PPT for re-screening. The ends of the left and right flanking sequence of the gene (*Cmozf*-up/down) (Table S1) were used as PCR primers to verify the transgenic lines. The selection method for overexpression transformants was similar to the above methods, except that PPT was replaced with sulfonylurea (Sur). Transformants were screened using PCR with primers *pb3*-F/*pb3*-OE*Cmozf*-R (Table S1), and qRT-PCR was used to determine the expression level of the target gene.

## Growth observation and conidial yield measurement

The PDA plate was plugged with a cork borer (1 cm in diameter). Sterile toothpicks were used to transfer inocula onto fresh PDA plates. Alternatively, the conidial concentration was adjusted to $1 \times 10^7$ conidia/mL, and 2 μL of the suspension was inoculated onto the PDA medium. The cultures were incubated in the dark at 25°C for 3 days and then exposed to light (200–300 lx) for 12 or 17 days before being photographed. The cultures were rinsed repeatedly with 3 mL of 0.05% Tween-80 to wash off conidia and then filtered through a double-layered lens cleaning paper to remove mycelia so as to obtain a conidial suspension. The number of conidia was then quantified using a hemocytometer.

## Scanning electron microscopy method

After the strains were inoculated on the PDA medium for 15 days, three mycelial blocks (1 cm in diameter) were selected using a cork borer and fixed in 2.5% glutaraldehyde at 4°C for 12–24 h. After the fixative was removed, the samples were rinsed three times with 0.1 M phosphate buffer (pH 7.2) for 15 min each time. Afterward, the samples were post-fixed with 1% osmium tetroxide for 1–2 h and re-rinsed three times with the same buffer. Dehydration was performed using a series of ethanol solutions (30%, 50%, 70%, 80%, 90%, and 95%), with each concentration applied for 15–20 min. Then, the dehydrated samples were treated with 100% ethanol for 20 min and immersed in fresh 100% ethanol. The samples were then dried using a carbon dioxide critical point dryer (Quorum K850, United Kingdom). Finally, the samples were mounted on a sample holder with conductive carbon tapes and sputter-coated with gold for 30 s using an ion sputter coater (Cressington 108 Auto, United Kingdom) for observation on a scanning electron microscope (Hitachi SU8010, Japan).

## Determination of carotenoid content

Three mycelial blocks were inoculated into the PDB medium and incubated at 25°C for 4 days under shaking at 150 r/min to obtain mycelium pellets. Subsequently, the cultures

were statically incubated at 25°C under illumination for 10 days. The mycelia were harvested, dried, and utilized for carotenoid determination as outlined by Zhao et al. (39). Briefly, 0.2 g of dried mycelia was cryogenically ground and dissolved in 70% ethanol at a ratio of 1:60 (material-to-solvent). The resulting mixture was sonicated at 418 W for 20 min and then extracted in a water bath at 50°C for 15 min. After centrifugation at 5,000 r/min for 10 min to remove the precipitate, absorbance was measured at 470 nm to calculate carotenoid content: carotenoid content ($\mu$g/g) = $(A \times V \times D)/(0.16 \times W)$, where $A$ represents absorbance, $V$ is the volume of the extracting solvent, $D$ is the dilution factor of the extract, 0.16 is the extinction coefficient, and $W$ is the dry weight of the sample (g).

## Determination of polysaccharide content

Polysaccharide was extracted from *C. militaris* using a modified hot water extraction method based on Xu et al. (40). Freeze-dried mycelia (0.1 g) were ground in liquid nitrogen and mixed with ddH$_2$O at a ratio of 1:40. The mixture was then heated in a water bath at 70°C for 4 h. After centrifugation at 10,000 r/min for 5 min, the supernatant was collected. Anhydrous ethanol was added to the supernatant to reach a concentration of 80% (vol/vol). Then, the solution was refrigerated at 4°C for 12 h to precipitate crude polysaccharides. The crude polysaccharides were dissolved in ddH$_2$O, and the pH was adjusted to 3.0 with 0.1 mol/L HCl and 4% (wt/vol) ZnSO$_4$. After overnight precipitation at 4°C and centrifugation at 10,000 r/min for 5 min, a purified polysaccharide solution was obtained. Hydrolysis was carried out by adding 6 mol/L HCl at a ratio of 1:4 and heating in a water bath at 100°C for 6 h. Next, the pH was neutralized to 7.0 with 6 mol/L NaOH and then centrifuged at 10,000 r/min for 5 min to collect the supernatant for further use. Polysaccharide content was determined using the 3,5-dinitrosalicylic acid method: polysaccharide (mg/g) = $(C \times V)/(m \times 1,000) \times$ dilution factor $\times 0.9$, where $C$ is the concentration of the hydrolyzed polysaccharide solution (mg/g), $V$ is the total volume, and $m$ is the weight of the mycelium.

## Real-time quantitative PCR

The strains were cultured in the dark on PDA medium for 3 days, followed by exposure to light at an intensity of 200–300 lx for 1, 4, and 7 days, or alternatively kept in the dark for a total of 7 days. Total RNA was isolated with the Eastep Super Total RNA Extraction Kit (Promega, USA) following the manufacturer's instructions, and then reverse transcription was carried out using a reverse transcription kit (Yugong Biotechnology, China). qRT-PCR was performed using SYBR Green qPCR Mix and qTOWER384G Real-Time PCR Detection System (Analytik Jena AG, Germany) according to the manufacturer's protocol. The PCR reaction system (10 $\mu$L) consisted of 5 $\mu$L of 2$\times$ SYBR Green qPCR mix, 0.2 $\mu$L (10 $\mu$M) of each of the primer pairs designed for the indicated genes (Table S1), 3 $\mu$L of diluted 20-fold cDNA template, and 1 $\mu$L of ddH$_2$O. The PCRs were performed as follows: initial denaturation at 95°C for 3 min; 40 cycles of 95°C for 15 s, 60°C for 15 s, and 72°C for 30 s. The comparative method of relative quantification (ddCt) was used to assess the relative quantification of gene expression using *Cmactin* as an internal standard.

## Transcriptome sequencing and bioinformatics analysis

Total RNA was extracted from strains subjected to 3-day dark incubation, followed by 4-day light exposure using TRIzol Reagent (Thermo Fisher Scientific, USA). RNA quality was assessed with a NanoDrop 2000 ($OD_{260/280}$: 1.8–2.2) and Agilent 5300 Bioanalyzer (RIN $\geq$ 6.5). Stranded mRNA libraries were constructed using the Illumina Stranded mRNA Prep, Ligation kit (1 $\mu$g input RNA) and sequenced on the NovaSeq X Plus platform (Illumina, USA) with 150 bp paired-end reads at Majorbio (Shanghai, China). Raw reads were quality-controlled using fastp (v0.23.4) to remove adapter sequences, low-quality bases (Q < 20), and reads with >10% N content or lengths <20 bp. Clean reads were aligned to the *Cordyceps militaris* CM01 reference genome using HISAT2 (v2.2.1) with default parameters. Gene expression quantification was performed via RSEM (v1.3.3)

and reported as transcripts per million. Differential expression analysis was conducted using DEGseq (v1.46.0) (adjusted *P* value [false discovery rate, FDR < 0.001]). Functional enrichment analysis of DEGs included gene ontology (GO) (Biological Process, Cellular Component, Molecular Function) and Kyoto Encyclopedia of Genes and Genomes pathway annotations, with significance determined by Fisher's exact test (FDR-corrected *P* < 0.05) using Goatools (v1.0.0) and SciPy (v1.7.1), respectively.

## Yeast one-hybrid assay

Yeast one-hybrid assays were performed following the protocol by Chen et al. (32). The *Cmozf* (1,425 bp) and *Cmwc-1* (2,892 bp) genes were amplified using specific primer pairs (Table S1) with *C. militaris* cDNA as the template. The amplified fragments were then inserted into the pB42AD vector to generate pB42AD-*Cmozf* or pB42AD-*Cmwc-1* (effector) constructs. Furthermore, the promoters of *CmbrlA* or *Cmwc-1* (~1,000 bp) were amplified from *C. militaris* genomic DNA and cloned into the pLacZi vector to produce reporter constructs. The effector and reporter constructs were co-transformed into the EGY48 yeast strain to investigate the interactions within yeast cells. The resulting transformants were grown on the SD/-Ura/-Trp medium at 30℃ for 3–5 days. Positive colonies were then transferred to the SD/Gal/Raf medium supplemented with X-gal (80 mg/L) to facilitate color development. Empty pB42AD and pLacZi vectors served as negative controls.

## ACKNOWLEDGMENTS

This study was supported by the Natural Science Foundation of Chongqing (CSTB2022NSCQ-MSX0261), the Scientific and Technological Research Program of Chongqing Municipal Education Commission (KJZD-K202201601), and the College Student Research Project of Chongqing University of Education (KY20230002).

## AUTHOR AFFILIATIONS

[1]College of Biological and Chemical Engineering, Chongqing University of Education, Chongqing, China
[2]The First Affiliated Hospital/Clinical Medical School Guangdong Pharmaceutical University, Guangzhou, China

## AUTHOR ORCIDs

Jin-feng Chen http://orcid.org/0000-0001-7867-7554

## FUNDING

| Funder | Grant(s) | Author(s) |
| --- | --- | --- |
| Natural Science Foundation of Chongqing Municipality | CSTB2022NSCQ-MSX0261 | Jinfeng Chen |
| Chongqing Municipal Education Commission | KJZD-K202201601 | Jinfeng Chen |

## AUTHOR CONTRIBUTIONS

Jin-feng Chen, Funding acquisition, Supervision, Writing – original draft | Fu-ling Cheng, Investigation | Tong-yue Chen, Investigation | Yi-lan Xu, Investigation | Jia-mei Song, Investigation | Hui-min Wang, Investigation | Yu Zhang, Investigation | Xi-chuan Guo, Software, Writing – review and editing | Jing Luo, Writing – review and editing

## DATA AVAILABILITY

The sequence data have been deposited in the Genome Sequence Archive (GSA) at the China National Center for Bioinformation (CNCB) under accession number CRA031452.

## ADDITIONAL FILES

The following material is available online.

## Supplemental Material

**Supplemental material (Spectrum01057-25-S0001.docx).** Fig. S1 to S7; Table S1.

## Open Peer Review

**PEER REVIEW HISTORY (review-history.pdf).** An accounting of the reviewer comments and feedback.

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
