## [Reviewer comments · Microbiology Spectrum]

Microbiology Spectrum

Light-responsive transcription factor CmOzf integrates conidiation, fruiting body development, and secondary metabolism in *Cordyceps militaris*

Jinfeng Chen, Fuling Cheng, Tongyue Chen, Yilan Xu, Jiamei Song, Huimin Wang, Yu Zhang, Xichuan Guo, and Jing Luo

Corresponding Author(s): Jinfeng Chen, College of Biological and Chemical Engineering, Chongqing University of Education, Chongqing 400067, China

Review Timeline:

Submission Date:	April 16, 2025
Editorial Decision:	May 14, 2025
Revision Received:	June 16, 2025
Editorial Decision:	July 9, 2025
Revision Received:	August 9, 2025
Editorial Decision:	August 22, 2025
Revision Received:	August 29, 2025
Accepted:	September 21, 2025

Editor: Jing Han

Reviewer(s): Disclosure of reviewer identity is with reference to reviewer comments included in decision letter(s). The following individuals involved in review of your submission have agreed to reveal their identity: Ye-Eun Son (Reviewer #1); Zhangxun Wang (Reviewer #4)

Transaction Report:

DOI: <https://doi.org/10.1128/spectrum.01057-25>

Re: Spectrum01057-25 (Light-responsive transcription factor CmOzf regulates conidial formation, fruiting development, and secondary metabolite production in *Cordyceps militaris*)

Dear Dr. Jinfeng chen:

Thank you for the privilege of reviewing your work. Below you will find my comments, instructions from the Spectrum editorial office, and the reviewer comments.

Revision Guidelines

Sincerely,
Jing Han
Editor
Microbiology Spectrum

Reviewer #2 (Comments for the Author):

The manuscript is well written and the data and experimental results are of good quality. The effect of CmOzf gene is studied and a range of possibly direct effects is described. I would suggest a couple of modifications. I would require to the authors to add details about the CmOzf protein. Readers would acknowledge a brief introduction on this gene and the reason for its selection in addition to increased expression under illumination and to which family of transcription

factor it belongs.

Secondly, I would suggest the improvement of figure 9. To improve its comprehension, I suggest the inclusion of graphs or diagrams explaining the basis of these functional assays in *S. cerevisiae*. In addition, I don't see any negative control for the expression of the CmOzf-AD protein. If CmOzf is a transcription factor binding the promoter sequences it is required as a control to show that other DNA sequence used with the reporter is not bound by this TF. Alternatively, introduce a mutation preventing DNA binding in the Ozf zinc finger region.

Remarks to the authors:

Chen et al. studied a light-responsive transcription factor, CmOzf, in *Cordyceps militaris*. They found that CmOzf regulates both conidial development and fruiting body formation in a light-dependent manner. Transcriptomic analysis and yeast one-hybrid assays indicated that CmOzf directly regulates Cmwc-1 and CmbrlA. These findings suggest that CmOzf modulates conidiation via *brlA* and contributes to the feedback inhibition of Cmwc-1, thereby linking light signaling to fungal development and secondary metabolism. However, the manuscript requires revision to improve scientific clarity and rigor, in order to meet the publication standards of *Microbiology Spectrum*.

1. Introduction: It seems that this study focuses on *ozf* as a homolog of BbSmr1, which has been shown in previous studies to play an important role in conidial development. It would strengthen the manuscript to provide more explanation about why *ozf* was selected for investigation and why it is considered significant in this context.

Additionally, if CmOzf is a previously uncharacterized gene, it might be helpful to briefly explain the reasoning behind naming it "*ozf*"—especially for readers who may not be familiar with its relationship to BbSmr1.

2. Did the authors extract RNA for qPCR or transcriptome analysis from cultures grown on PDA solid media? If so, were the samples a mixture of conidia and hyphae? It would be helpful to clarify the developmental stage or cell type used for RNA extraction, as this could influence gene expression profiles.

In addition, for the transcriptome analysis, it would improve the clarity of the Methods section to include more detailed information on the raw data processing—specifically, which tools and pipelines were used (e.g., for quality control, trimming, alignment, normalization, etc.).

3. In the central conidiation pathway, different regulators function at distinct time points to control conidiophore formation and conidial maturation. Would it be possible to examine the time-course expression of *brlA*, *abaA*, and *wetA* in WT, *ozf*, and *wc1* mutant strains? Including such data could provide valuable insights into how each gene functions at different developmental stages. It might also enhance the scientific value of Figure 10.

Minor:

- Importance: BbOzf ... CmOzf
- Line 138, C2H2 ... C₂H₂
- Line 319, experiments ... studies
- Line 441, ZnSO4 ... ZnSO₄
- Line 519, 和 ... ?
- Line 544-545, *Bb* ... *Cm*
- Line 696: “,” ... “:”
- Figure 5A, Dolysaccharide ... Polysaccharide

Prof. Jing Han

Editor

Microbiology Spectrum

We are grateful to you and the two reviewers for the constructive comments on our manuscript entitled "Light-responsive transcription factor CmOzf regulates conidial formation, fruiting development, and secondary metabolite production in *Cordyceps militaris*". The title has been modified to "Light-responsive transcription factor CmOzf integrates conidiation, fruiting body development, and secondary metabolism in *Cordyceps militaris*" to better reflect the main aspect of the paper. The manuscript has been revised based on the Editorial and Reviewers' suggestions. In particular, as per the reviewers' comments additional data has been added including examining the time-course expression of *brlA*, *abaA*, and *wetA* in WT, $\Delta Cmozf$, and $\Delta Cmwcl$ mutant strains. Using *Volvariella volvacea Vvcbf* promoter as a negative control to show that other DNA sequence is not bound by CmOzf. Introduction, Results, Discussion, and Materials and Methods have been modified/updated to include the new data as well as in response to specific reviewers' comments. The language has been improved with the help of American Journal Experts. We feel that the quality of our paper has been substantially improved after revision and appreciate your consideration of this work.

The complete reviewers' comments and a point-by-point responses are included in the revised submission.

Sincerely yours,

Jinfeng Chen

Reviewer: 1

Comments to the Author

The manuscript is well written and the data and experimental results are of good quality. The effect of CmOzf gene is studied and a range of possibly direct effects is described. I would suggest a couple of modifications.

Major concerns:

1. I would require to the authors to add details about the CmOzf protein. Readers would acknowledge a brief introduction on this gene and the reason for its selection in addition to increased expression under illumination and to which family of transcription factor it belongs.

Response: We sincerely appreciate the reviewer's constructive suggestion. As requested, we have now added a detailed introduction to CmOzf in the Introduction and Results section (Page 7, Line 126-132; Page8, Line 136-140), including:

Classification: CmOzf belongs to the C₂H₂-type zinc finger transcription factor family, homologous to BbSmr1 in *Beauveria bassiana*.

Functional rationale: Its selection was based on (1) light-responsive expression patterns linked to carotenoid biosynthesis and conidiation, and (2) its structural homology with BbSmr1, a known regulator of conidial development and secondary metabolism.

Regulatory role: We emphasized its potential as a bridge between light signaling (via CmWC-1) and developmental pathways (BrlA-AbaA-WetA).

2. Secondly, I would suggest the improvement of figure 9. To improve its comprehension, I suggest the inclusion of graphs or diagrams explaining the basis of these functional assays in *S. cerevisiae*. In addition, I don't see any negative control for the expression of the CmOzf-AD protein. If CmOzf is a transcription factor binding the promoter sequences it is required as a control to show that other DNA sequence used with the reporter is not bound by this TF. Alternatively, introduce a mutation preventing DNA binding in the Ozf zinc finger region.

Response: We sincerely appreciate the reviewer's valuable suggestions for enhancing Figure 9. In response, we have added a new schematic diagram (Fig S6-B) that clearly illustrates: (1) the experimental design of our yeast one-hybrid assays in *S. cerevisiae*, (2) the system configuration and working mechanism, (3) the specific interaction between the transcription factor (TF) and target promoter, and (4) the expected transcriptional activation outcomes. This addition significantly improves the clarity and interpretability of our functional assay results.

2. We have performed additional control experiments as suggested: we included an experiment with a scrambled *Vvcbf* promoter sequence that doesn't contain the predicted CmOzf binding sites (Fig. S6-A), demonstrating no activation of the reporter system.

Reviewer: 2

Comments to the Author

Chen et al. studied a light-responsive transcription factor, CmOzf, in *Cordyceps militaris*. They found that CmOzf regulates both conidial development and fruiting body formation in a light-dependent manner. Transcriptomic analysis and yeast one-hybrid assays indicated that CmOzf directly regulates Cmwc-1 and CmbrlA. These findings suggest that CmOzf modulates conidiation via brlA and contributes to the feedback inhibition of Cmwc-1, thereby linking light signaling to fungal development and secondary metabolism. However, the manuscript requires

revision to improve scientific clarity and rigor, in order to meet the publication standards of Microbiology Spectrum. Major concerns:

1. Introduction: It seems that this study focuses on *ozf* as a homolog of *BbSmr1*, which has been shown in previous studies to play an important role in conidial development. It would strengthen the manuscript to provide more explanation about why *ozf* was selected for investigation and why it is considered significant in this context. Additionally, if *CmOzf* is a previously uncharacterized gene, it might be helpful to briefly explain the reasoning behind naming it "*ozf*"—especially for readers who may not be familiar with its relationship to *BbSmr1*.

Response: We sincerely appreciate the reviewer's insightful suggestion regarding the rationale for studying *CmOzf*. In response, we have expanded the Introduction section (Page 7, Lines 126-132) to provide: (1) clear justification for selecting *CmOzf* based on its significant homology with *BbSmr1* (a characterized conidiation regulator in *Beauveria bassiana* [16]) and the presence of a conserved C₂H₂ zinc finger domain indicative of DNA-binding activity; (2) explanation of the nomenclature "*ozf*" (orange pigmentation-related zinc finger protein) reflecting both its structural features and observed role in secondary metabolism regulation; and (3) supporting evidence from our preliminary data showing that *CmOzf* knockout strains exhibit not only defective conidiation but also altered orange pigmentation (Results Section 2). These modifications significantly strengthen the biological context and research justification for *CmOzf* investigation.

2. Did the authors extract RNA for qPCR or transcriptome analysis from cultures grown on PDA solid media? If so, were the samples a mixture of conidia and hyphae? It would be helpful to clarify the developmental stage or cell type used for RNA extraction, as this could influence gene expression profiles. In addition, for the transcriptome analysis, it would improve the clarity of the Methods section to include more detailed information on the raw data processing—specifically, which tools and pipelines were used (e.g., for quality control, trimming, alignment, normalization, etc.).

Response: We appreciate the reviewer's insightful question regarding our RNA source selection. Our choice to use the fungi rather than conidia for RNA extraction was based on both biological and technical considerations: (1) As *CmBrlA* functions as a key initiator of conidiophore development during hyphal differentiation (prior to conidium formation), analyzing its expression in mature conidia would not reflect its primary regulatory role; (2) Practical attempts to extract RNA from the scarce conidia produced by $\Delta Cmwc-1$ and $\Delta Cmofz$ mutants (about 10% of wild-type yield) yielded insufficient quantities (<20 ng/ μ L) and inconsistent qPCR results, prompting us to abandon this approach; (3) Mycelial RNA better represents the natural physiological state during the critical developmental transition when *CmbrlA* is functionally active, while also providing more reliable and reproducible data.

We have now expanded the Methods section to detail our transcriptome analysis pipeline: (1) Total RNA was extracted from fungal strains after 3-day dark/4-day light treatment using TRIzol® Reagent (Thermo Fisher Scientific), with quality verification by NanoDrop2000 (OD_{260/280}=1.8-2.2) and Agilent 5300 Bioanalyzer (RIN \geq 6.5). Stranded mRNA libraries were prepared using Illumina® Stranded mRNA Prep kit (1 μ g input) and sequenced on NovaSeq X Plus (150 bp PE reads) at Majorbio (Shanghai); (2) Raw reads were processed with fastp (v0.23.4) to remove adapters, low-quality bases (Q<20), reads with >10% N content, and fragments <20 bp; (3) Clean reads were aligned to the *Cordyceps militaris* CM01 genome using HISAT2 (v2.2.1) with default parameters; (4) Gene expression quantification was performed via RSEM (v1.3.3)

and normalized as TPM values; (5) Differential expression analysis used DEGseq (v1.46.0) with thresholds of $|\log_2FC| \geq 1$ and $FDR < 0.001$; (6) Functional enrichment of DEGs included GO term (Biological Process/Cellular Component/Molecular Function) and KEGG pathway analyses, conducted with Goatools (v1.0.0) and SciPy (v1.7.1) respectively using Fisher's exact test (FDR-corrected $p < 0.05$). Complete parameters are now provided in Page 26-27, Line 492-512.

3. In the central conidiation pathway, different regulators function at distinct time points to control conidiophore formation and conidial maturation. Would it be possible to examine the time-course expression of *brlA*, *abaA*, and *wetA* in WT, *ozf*, and *wc1* mutant strains? Including such data could provide valuable insights into how each gene functions at different developmental stages. It might also enhance the scientific value of Figure 10.

Response: We sincerely appreciate the reviewer's insightful suggestion. To investigate the temporal regulation of the central conidiation pathway, we performed time-course qPCR analyses of *CmbrlA*, *CmabaA*, and *CmwetA* in WT, $\Delta Cmzf$, and $\Delta Cmwc-1$ strains during conidial development (4 - 10 days, sampled at 3-day intervals). Key findings include:

(1) WT Strain: *CmbrlA*, *CmabaA*, and *CmwetA* exhibited synchronized peak expression at day 7, followed by gradual decline (Fig. 10A - C).

(2) $\Delta Cmzf$ mutant: Significant downregulation of *CmbrlA*, *CmabaA*, and *CmwetA* at all time points (Fig. S7-A). *Cmwc-1* expression was significantly upregulated at days 7 and 10, suggesting feedback regulation.

(3) $\Delta Cmwc-1$ mutant: *CmbrlA* showed delayed induction (reduced at day 4 but upregulated at days 7 and 10). *Cmzf* expression was markedly suppressed at all stages (Fig. S7-B), indicating cross-regulation between *Cmwc-1* and *Cmzf*.

These results (now integrated into Fig. 10 and Supplementary Fig. S7, Page 14-15, Line 254-269) reveal stage-specific disruptions in the conidiation cascade and highlight complex genetic interactions between *CmWc-1*, *CmOzf*, and the core developmental regulators. We believe these data significantly enhance the mechanistic understanding of conidiation control in our study.

4. Minor: - - - - - Importance: BbOzf ... CmOzf Line 138, C2H2 ... C2H2 Line 319, experiments ... studies Line 441, ZnSO4 ... ZnSO4 Line 519, 和 ... ? Line 544-545, Bb ... Cm Line 696: “,” ... “:” Figure 5A, Dolysaccharide ... Polysaccharide

Response: The detailed contents had been revised accordingly throughout the manuscript.

Re: Spectrum01057-25R1 (Light-responsive transcription factor CmOzf integrates conidiation, fruiting body development, and secondary metabolism in *Cordyceps militaris*)

Dear Dr. Jinfeng Chen:

Thank you for the privilege of reviewing your work. Below you will find my comments, instructions from the Spectrum editorial office, and the reviewer comments.

Revision Guidelines

Sincerely,
Jing Han
Editor
Microbiology Spectrum

Reviewer #2 (Comments for the Author):

The authors have addressed all my queries.

Reviewer #3 (Comments for the Author):

In this revised version, the authors have addressed some of the previous reviewer comments. However, several important issues remain.

- Line 21: Remove "(*C. militaris*)".
- Line 111: Replace "Flug" with "FluG".
- Line 145: Please check the mRNA expression level of the *Cmozf* gene. The reported 2.5-fold increase may not be accurate.
- Lines 180-181: Please check the fold-change values; they appear unclear or inconsistent.
- Figure 6B: Separate the up-regulated and down-regulated genes in the diagram. Then, perform GO analysis based on this classification. At present, it is difficult to draw clear conclusions from the KEGG or GO analysis.
- Figure 8: Include the phenotype of the OECmozf strain in this figure.
- Figure 10: The conclusion that WC-1 represses *brlA* expression is not clearly supported by the data. Similarly, the claim that *Ozf* increases *abaA* expression is also unclear.
- *AbaA* cannot be simply described as a gene involved in the conidial maturation pathway. Please revise this interpretation.
- Figure 2C: Clearly indicate the spores using arrows for better visualization.
- Figures (general): Please align text and images uniformly across all figures to improve visual clarity and presentation quality.

Prof. Jing Han

Editor

Microbiology Spectrum

We are grateful to you and the two reviewers for the constructive comments on our manuscript entitled "Light-responsive transcription factor CmOzf regulates conidial formation, fruiting development, and secondary metabolite production in *Cordyceps militaris*". The title has been modified to "Light-responsive transcription factor CmOzf integrates conidiation, fruiting body development, and secondary metabolism in *Cordyceps militaris*" to better reflect the main aspect of the paper. The manuscript has been revised based on the Editorial and Reviewers' suggestions. In particular, as per the reviewers' comments additional data has been added including examining the time-course expression of *brlA*, *abaA*, and *wetA* in WT, $\Delta Cmozf$, and $\Delta Cmwcl$ mutant strains. Using *Volvariella volvacea Vvcbf* promoter as a negative control to show that other DNA sequence is not bound by CmOzf. Introduction, Results, Discussion, and Materials and Methods have been modified/updated to include the new data as well as in response to specific reviewers' comments. The language has been improved with the help of American Journal Experts. We feel that the quality of our paper has been substantially improved after revision and appreciate your consideration of this work.

The complete reviewers' comments and a point-by-point responses are included in the revised submission.

Sincerely yours,

Jinfeng Chen

Reviewer: 1

Comments to the Author

The manuscript is well written and the data and experimental results are of good quality. The effect of CmOzf gene is studied and a range of possibly direct effects is described. I would suggest a couple of modifications.

Major concerns:

1. I would require to the authors to add details about the CmOzf protein. Readers would acknowledge a brief introduction on this gene and the reason for its selection in addition to increased expression under illumination and to which family of transcription factor it belongs.

Response: We sincerely appreciate the reviewer's constructive suggestion. As requested, we have now added a detailed introduction to CmOzf in the Introduction and Results section (Page 7, Line 126-132; Page 8, Line 136-140), including:

Classification: CmOzf belongs to the C₂H₂-type zinc finger transcription factor family, homologous to BbSmr1 in *Beauveria bassiana*.

Functional rationale: Its selection was based on (1) light-responsive expression patterns linked to carotenoid biosynthesis and conidiation, and (2) its structural homology with BbSmr1, a known regulator of conidial development and secondary metabolism.

Regulatory role: We emphasized its potential as a bridge between light signaling (via CmWC-1) and developmental pathways (BrlA-AbaA-WetA).

2. Secondly, I would suggest the improvement of figure 9. To improve its comprehension, I suggest the inclusion of graphs or diagrams explaining the basis of these functional assays in *S. cerevisiae*. In addition, I don't see any negative control for the expression of the CmOzf-AD protein. If CmOzf is a transcription factor binding the promoter sequences it is required as a control to show that other DNA sequence used with the reporter is not bound by this TF. Alternatively, introduce a mutation preventing DNA binding in the Ozf zinc finger region.

Response: We sincerely appreciate the reviewer's valuable suggestions for enhancing Figure 9. In response, we have added a new schematic diagram (Fig S6-B) that clearly illustrates: (1) the experimental design of our yeast one-hybrid assays in *S. cerevisiae*, (2) the system configuration and working mechanism, (3) the specific interaction between the transcription factor (TF) and target promoter, and (4) the expected transcriptional activation outcomes. This addition significantly improves the clarity and interpretability of our functional assay results.

2. We have performed additional control experiments as suggested: we included an experiment with a scrambled *Vvcbf* promoter sequence that doesn't contain the predicted CmOzf binding sites (Fig. S6-A), demonstrating no activation of the reporter system.

Reviewer: 2

Comments to the Author

Chen et al. studied a light-responsive transcription factor, CmOzf, in *Cordyceps militaris*. They found that CmOzf regulates both conidial development and fruiting body formation in a light-dependent manner. Transcriptomic analysis and yeast one-hybrid assays indicated that CmOzf directly regulates Cmwc-1 and CmbrlA. These findings suggest that CmOzf modulates conidiation via brlA and contributes to the feedback inhibition of Cmwc-1, thereby linking light signaling to fungal development and secondary metabolism. However, the manuscript requires

revision to improve scientific clarity and rigor, in order to meet the publication standards of Microbiology Spectrum. Major concerns:

1. Introduction: It seems that this study focuses on *ozf* as a homolog of *BbSmr1*, which has been shown in previous studies to play an important role in conidial development. It would strengthen the manuscript to provide more explanation about why *ozf* was selected for investigation and why it is considered significant in this context. Additionally, if *CmOzf* is a previously uncharacterized gene, it might be helpful to briefly explain the reasoning behind naming it "*ozf*"—especially for readers who may not be familiar with its relationship to *BbSmr1*.

Response: We sincerely appreciate the reviewer's insightful suggestion regarding the rationale for studying *CmOzf*. In response, we have expanded the Introduction section (Page 7, Lines 126-132) to provide: (1) clear justification for selecting *CmOzf* based on its significant homology with *BbSmr1* (a characterized conidiation regulator in *Beauveria bassiana* [16]) and the presence of a conserved C₂H₂ zinc finger domain indicative of DNA-binding activity; (2) explanation of the nomenclature "*ozf*" (orange pigmentation-related zinc finger protein) reflecting both its structural features and observed role in secondary metabolism regulation; and (3) supporting evidence from our preliminary data showing that *CmOzf* knockout strains exhibit not only defective conidiation but also altered orange pigmentation (Results Section 2). These modifications significantly strengthen the biological context and research justification for *CmOzf* investigation.

2. Did the authors extract RNA for qPCR or transcriptome analysis from cultures grown on PDA solid media? If so, were the samples a mixture of conidia and hyphae? It would be helpful to clarify the developmental stage or cell type used for RNA extraction, as this could influence gene expression profiles. In addition, for the transcriptome analysis, it would improve the clarity of the Methods section to include more detailed information on the raw data processing—specifically, which tools and pipelines were used (e.g., for quality control, trimming, alignment, normalization, etc.).

Response: We appreciate the reviewer's insightful question regarding our RNA source selection. Our choice to use the fungi rather than conidia for RNA extraction was based on both biological and technical considerations: (1) As *CmBrlA* functions as a key initiator of conidiophore development during hyphal differentiation (prior to conidium formation), analyzing its expression in mature conidia would not reflect its primary regulatory role; (2) Practical attempts to extract RNA from the scarce conidia produced by $\Delta Cmwc-1$ and $\Delta Cmofz$ mutants (about 10% of wild-type yield) yielded insufficient quantities (<20 ng/ μ L) and inconsistent qPCR results, prompting us to abandon this approach; (3) Mycelial RNA better represents the natural physiological state during the critical developmental transition when *CmbrlA* is functionally active, while also providing more reliable and reproducible data.

We have now expanded the Methods section to detail our transcriptome analysis pipeline: (1) Total RNA was extracted from fungal strains after 3-day dark/4-day light treatment using TRIzol® Reagent (Thermo Fisher Scientific), with quality verification by NanoDrop2000 (OD_{260/280}=1.8-2.2) and Agilent 5300 Bioanalyzer (RIN \geq 6.5). Stranded mRNA libraries were prepared using Illumina® Stranded mRNA Prep kit (1 μ g input) and sequenced on NovaSeq X Plus (150 bp PE reads) at Majorbio (Shanghai); (2) Raw reads were processed with fastp (v0.23.4) to remove adapters, low-quality bases (Q<20), reads with >10% N content, and fragments <20 bp; (3) Clean reads were aligned to the *Cordyceps militaris* CM01 genome using HISAT2 (v2.2.1) with default parameters; (4) Gene expression quantification was performed via RSEM (v1.3.3)

and normalized as TPM values; (5) Differential expression analysis used DEGseq (v1.46.0) with thresholds of $|\log_2FC| \geq 1$ and $FDR < 0.001$; (6) Functional enrichment of DEGs included GO term (Biological Process/Cellular Component/Molecular Function) and KEGG pathway analyses, conducted with Goatools (v1.0.0) and SciPy (v1.7.1) respectively using Fisher's exact test (FDR-corrected $p < 0.05$). Complete parameters are now provided in Page 26-27, Line 492-512.

3. In the central conidiation pathway, different regulators function at distinct time points to control conidiophore formation and conidial maturation. Would it be possible to examine the time-course expression of *brlA*, *abaA*, and *wetA* in WT, *ozf*, and *wc1* mutant strains? Including such data could provide valuable insights into how each gene functions at different developmental stages. It might also enhance the scientific value of Figure 10.

Response: We sincerely appreciate the reviewer's insightful suggestion. To investigate the temporal regulation of the central conidiation pathway, we performed time-course qPCR analyses of *CmbrlA*, *CmabaA*, and *CmwetA* in WT, $\Delta Cmofz$, and $\Delta Cmwc-1$ strains during conidial development (4 - 10 days, sampled at 3-day intervals). Key findings include:

(1) WT Strain: *CmbrlA*, *CmabaA*, and *CmwetA* exhibited synchronized peak expression at day 7, followed by gradual decline (Fig. 10A - C).

(2) $\Delta Cmofz$ mutant: Significant downregulation of *CmbrlA*, *CmabaA*, and *CmwetA* at all time points (Fig. S7-A). *Cmwc-1* expression was significantly upregulated at days 7 and 10, suggesting feedback regulation.

(3) $\Delta Cmwc-1$ mutant: *CmbrlA* showed delayed induction (reduced at day 4 but upregulated at days 7 and 10). *Cmofz* expression was markedly suppressed at all stages (Fig. S7-B), indicating cross-regulation between *Cmwc-1* and *Cmofz*.

These results (now integrated into Fig. 10 and Supplementary Fig. S7, Page 14-15, Line 254-269) reveal stage-specific disruptions in the conidiation cascade and highlight complex genetic interactions between *CmWc-1*, *CmOzf*, and the core developmental regulators. We believe these data significantly enhance the mechanistic understanding of conidiation control in our study.

4. Minor: - - - - - Importance: *BbOzf* ... *CmOzf* Line 138, *C2H2* ... *C2H2* Line 319, experiments ... studies Line 441, *ZnSO4* ... *ZnSO4* Line 519, 和 ... ? Line 544-545, *Bb* ... *Cm* Line 696: “,” ... “:” Figure 5A, Dolysaccharide ... Polysaccharide

Response: The detailed contents had been revised accordingly throughout the manuscript.

Prof. Jing Han

Editor

Microbiology Spectrum

We are grateful to you and the reviewers for the constructive comments on our manuscript entitled "Light-responsive transcription factor CmOzf integrates conidiation, fruiting body development, and secondary metabolism in *Cordyceps militaris*". The manuscript has been revised based on the Editorial and Reviewers' suggestions. Introduction, Results, Discussion, and Materials and Methods have been modified/updated to include the new data as well as in response to specific reviewers' comments. We feel that the quality of our paper has been substantially improved after revision and appreciate your consideration of this work.

The complete reviewers' comments and a point-by-point responses are included in the revised submission.

Sincerely yours,

Jinfeng Chen

Reviewer: 3

In this revised version, the authors have addressed some of the previous reviewer comments. However, several important issues remain.

- Line 21: Remove "(C. militaris)".

Response: In line 21, we have removed "(C. militaris)".

- Line 111: Replace "Flug" with "FluG".

Response: Line 111: we have replace "Flug" with "FluG".

- Line 145: Please check the mRNA expression level of the *Cmozf* gene. The reported 2.5-fold increase may not be accurate.

Response: Line 145: The mRNA expression level of *Cmozf* was corrected to 3.5-fold (revised from 2.5-fold).

- Lines 180-181: Please check the fold-change values; they appear unclear or inconsistent.

Response: Lines 180-181 and Lines 184-185, the fold-change values were all revised.

- Figure 6B: Separate the up-regulated and down-regulated genes in the diagram.

Then, perform GO analysis based on this classification. At present, it is difficult to draw clear conclusions from the KEGG or GO analysis.

Response: Lines 197-206: Venn analysis was performed separately for the up-regulated and down-regulated genes, as shown in Figure 6B. Significantly up-regulated gene sets and significantly down-regulated gene sets identified from Figure 6B were subjected to independent Gene Ontology (GO) enrichment analyses, as illustrated in Figure 6C.

- Figure 8: Include the phenotype of the OEC*Cmozf* strain in this figure.

Response: Lines 221-223 and 229-232: The phenotype of the OEC*Cmozf* strain and

its associated gene expression profiles have been incorporated into Figure 8.

- Figure 10: The conclusion that WC-1 represses brlA expression is not clearly supported by the data. Similarly, the claim that Ozf increases abaA expression is also unclear.

- AbaA cannot be simply described as a gene involved in the conidial maturation pathway. Please revise this interpretation.

Response: Lines 254-258: Revised the gene expression experimental results to ensure data consistency and accuracy in the manuscript; removed the uncertain pathways of Ozf-AbaA and WC-1-BrlA based on experimental findings; redesigned the regulatory pathway of Ozf according to the results, as shown in Fig 10-B.

- Figure 2C: Clearly indicate the spores using arrows for better visualization.

Response: White arrows have been added in Figure 2-C to indicate the location of conidia and enhance visual identification.

- Figures (general): Please align text and images uniformly across all figures to improve visual clarity and presentation quality.

Response: The figures and text descriptions have been rechecked and revised to ensure consistency.

Re: Spectrum01057-25R2 (Light-responsive transcription factor CmOzf integrates conidiation, fruiting body development, and secondary metabolism in *Cordyceps militaris*)

Dear Dr. Jinfeng Chen:

Thank you for the privilege of reviewing your work. Below you will find my comments, instructions from the Spectrum editorial office, and the reviewer comments.

Revision Guidelines

Sincerely,
Jing Han
Editor
Microbiology Spectrum

Reviewer #3 (Comments for the Author):

I have no additional comments for this manuscript.

Reviewer #4 (Comments for the Author):

There are just responses to the one reviewer comments, and no response to this comments:

Specific comments:

1. The sentence "Light signal significantly affected ... (Fig. 1)." should be followed the subtitle "Cmozf is highly expressed under light exposure."
2. The mutants were cultured in the dark on PDA medium for 3 days, followed by 4 days of light exposure (Line 192-194), the authors should explain why the time point (4 days of light exposure) was selected for RNA-seq, and also provide the time-course expression profiles of Cmw-1 in WT (for clarifying the response of Cmw-1 to light).
3. For overexpression strains, the authors should make sure which one was used for phenotypic assay (Line 223-226), just like in gene disruption strains.
4. The label of Δ should be unified in all manuscript.
5. X axis in Figure 6 C and D should be the same (Line 564-569), please revised it.

Prof. Jing Han

Editor

Microbiology Spectrum

We are grateful to you and the two reviewers for the constructive comments on our manuscript entitled "Light-responsive transcription factor CmOzf regulates conidial formation, fruiting development, and secondary metabolite production in *Cordyceps militaris*". The title has been modified to "Light-responsive transcription factor CmOzf integrates conidiation, fruiting body development, and secondary metabolism in *Cordyceps militaris*" to better reflect the main aspect of the paper. The manuscript has been revised based on the Editorial and Reviewers' suggestions. In particular, as per the reviewers' comments additional data has been added including examining the time-course expression of *brlA*, *abaA*, and *wetA* in WT, $\Delta Cmozf$, and $\Delta Cmwcl$ mutant strains. Using *Volvariella volvacea Vvcbf* promoter as a negative control to show that other DNA sequence is not bound by CmOzf. Introduction, Results, Discussion, and Materials and Methods have been modified/updated to include the new data as well as in response to specific reviewers' comments. The language has been improved with the help of American Journal Experts. We feel that the quality of our paper has been substantially improved after revision and appreciate your consideration of this work.

The complete reviewers' comments and a point-by-point responses are included in the revised submission.

Sincerely yours,

Jinfeng Chen

Reviewer: 1

Comments to the Author

The manuscript is well written and the data and experimental results are of good quality. The effect of CmOzf gene is studied and a range of possibly direct effects is described. I would suggest a couple of modifications.

Major concerns:

1. I would require to the authors to add details about the CmOzf protein. Readers would acknowledge a brief introduction on this gene and the reason for its selection in addition to increased expression under illumination and to which family of transcription factor it belongs.

Response: We sincerely appreciate the reviewer's constructive suggestion. As requested, we have now added a detailed introduction to CmOzf in the Introduction and Results section (Page 7, Line 126-132; Page 8, Line 136-140), including:

Classification: CmOzf belongs to the C₂H₂-type zinc finger transcription factor family, homologous to BbSmr1 in *Beauveria bassiana*.

Functional rationale: Its selection was based on (1) light-responsive expression patterns linked to carotenoid biosynthesis and conidiation, and (2) its structural homology with BbSmr1, a known regulator of conidial development and secondary metabolism.

Regulatory role: We emphasized its potential as a bridge between light signaling (via CmWC-1) and developmental pathways (BrlA-AbaA-WetA).

2. Secondly, I would suggest the improvement of figure 9. To improve its comprehension, I suggest the inclusion of graphs or diagrams explaining the basis of these functional assays in *S. cerevisiae*. In addition, I don't see any negative control for the expression of the CmOzf-AD protein. If CmOzf is a transcription factor binding the promoter sequences it is required as a control to show that other DNA sequence used with the reporter is not bound by this TF. Alternatively, introduce a mutation preventing DNA binding in the Ozf zinc finger region.

Response: We sincerely appreciate the reviewer's valuable suggestions for enhancing Figure 9. In response, we have added a new schematic diagram (Fig S6-B) that clearly illustrates: (1) the experimental design of our yeast one-hybrid assays in *S. cerevisiae*, (2) the system configuration and working mechanism, (3) the specific interaction between the transcription factor (TF) and target promoter, and (4) the expected transcriptional activation outcomes. This addition significantly improves the clarity and interpretability of our functional assay results.

2. We have performed additional control experiments as suggested: we included an experiment with a scrambled *Vvcbf* promoter sequence that doesn't contain the predicted CmOzf binding sites (Fig. S6-A), demonstrating no activation of the reporter system.

Reviewer: 2

Comments to the Author

Chen et al. studied a light-responsive transcription factor, CmOzf, in *Cordyceps militaris*. They found that CmOzf regulates both conidial development and fruiting body formation in a light-dependent manner. Transcriptomic analysis and yeast one-hybrid assays indicated that CmOzf directly regulates *Cmwc-1* and *CmbrlA*. These findings suggest that CmOzf modulates conidiation via *brlA* and contributes to the feedback inhibition of *Cmwc-1*, thereby linking light signaling to fungal development and secondary metabolism. However, the manuscript requires

revision to improve scientific clarity and rigor, in order to meet the publication standards of Microbiology Spectrum. Major concerns:

1. Introduction: It seems that this study focuses on *ozf* as a homolog of *BbSmr1*, which has been shown in previous studies to play an important role in conidial development. It would strengthen the manuscript to provide more explanation about why *ozf* was selected for investigation and why it is considered significant in this context. Additionally, if *CmOzf* is a previously uncharacterized gene, it might be helpful to briefly explain the reasoning behind naming it "*ozf*"—especially for readers who may not be familiar with its relationship to *BbSmr1*.

Response: We sincerely appreciate the reviewer's insightful suggestion regarding the rationale for studying *CmOzf*. In response, we have expanded the Introduction section (Page 7, Lines 126-132) to provide: (1) clear justification for selecting *CmOzf* based on its significant homology with *BbSmr1* (a characterized conidiation regulator in *Beauveria bassiana* [16]) and the presence of a conserved C₂H₂ zinc finger domain indicative of DNA-binding activity; (2) explanation of the nomenclature "*ozf*" (orange pigmentation-related zinc finger protein) reflecting both its structural features and observed role in secondary metabolism regulation; and (3) supporting evidence from our preliminary data showing that *CmOzf* knockout strains exhibit not only defective conidiation but also altered orange pigmentation (Results Section 2). These modifications significantly strengthen the biological context and research justification for *CmOzf* investigation.

2. Did the authors extract RNA for qPCR or transcriptome analysis from cultures grown on PDA solid media? If so, were the samples a mixture of conidia and hyphae? It would be helpful to clarify the developmental stage or cell type used for RNA extraction, as this could influence gene expression profiles. In addition, for the transcriptome analysis, it would improve the clarity of the Methods section to include more detailed information on the raw data processing—specifically, which tools and pipelines were used (e.g., for quality control, trimming, alignment, normalization, etc.).

Response: We appreciate the reviewer's insightful question regarding our RNA source selection. Our choice to use the fungi rather than conidia for RNA extraction was based on both biological and technical considerations: (1) As *CmBrlA* functions as a key initiator of conidiophore development during hyphal differentiation (prior to conidium formation), analyzing its expression in mature conidia would not reflect its primary regulatory role; (2) Practical attempts to extract RNA from the scarce conidia produced by $\Delta Cmwc-1$ and $\Delta Cmofz$ mutants (about 10% of wild-type yield) yielded insufficient quantities (<20 ng/ μ L) and inconsistent qPCR results, prompting us to abandon this approach; (3) Mycelial RNA better represents the natural physiological state during the critical developmental transition when *CmbrlA* is functionally active, while also providing more reliable and reproducible data.

We have now expanded the Methods section to detail our transcriptome analysis pipeline: (1) Total RNA was extracted from fungal strains after 3-day dark/4-day light treatment using TRIzol® Reagent (Thermo Fisher Scientific), with quality verification by NanoDrop2000 (OD_{260/280}=1.8-2.2) and Agilent 5300 Bioanalyzer (RIN \geq 6.5). Stranded mRNA libraries were prepared using Illumina® Stranded mRNA Prep kit (1 μ g input) and sequenced on NovaSeq X Plus (150 bp PE reads) at Majorbio (Shanghai); (2) Raw reads were processed with fastp (v0.23.4) to remove adapters, low-quality bases (Q<20), reads with >10% N content, and fragments <20 bp; (3) Clean reads were aligned to the *Cordyceps militaris* CM01 genome using HISAT2 (v2.2.1) with default parameters; (4) Gene expression quantification was performed via RSEM (v1.3.3)

and normalized as TPM values; (5) Differential expression analysis used DEGseq (v1.46.0) with thresholds of $|\log_2FC| \geq 1$ and $FDR < 0.001$; (6) Functional enrichment of DEGs included GO term (Biological Process/Cellular Component/Molecular Function) and KEGG pathway analyses, conducted with Goatools (v1.0.0) and SciPy (v1.7.1) respectively using Fisher's exact test (FDR-corrected $p < 0.05$). Complete parameters are now provided in Page 26-27, Line 492-512.

3. In the central conidiation pathway, different regulators function at distinct time points to control conidiophore formation and conidial maturation. Would it be possible to examine the time-course expression of *brlA*, *abaA*, and *wetA* in WT, *ozf*, and *wc1* mutant strains? Including such data could provide valuable insights into how each gene functions at different developmental stages. It might also enhance the scientific value of Figure 10.

Response: We sincerely appreciate the reviewer's insightful suggestion. To investigate the temporal regulation of the central conidiation pathway, we performed time-course qPCR analyses of *CmbrlA*, *CmabaA*, and *CmwetA* in WT, $\Delta Cmzf$, and $\Delta Cmwc-1$ strains during conidial development (4 - 10 days, sampled at 3-day intervals). Key findings include:

(1) WT Strain: *CmbrlA*, *CmabaA*, and *CmwetA* exhibited synchronized peak expression at day 7, followed by gradual decline (Fig. 10A - C).

(2) $\Delta Cmzf$ mutant: Significant downregulation of *CmbrlA*, *CmabaA*, and *CmwetA* at all time points (Fig. S7-A). *Cmwc-1* expression was significantly upregulated at days 7 and 10, suggesting feedback regulation.

(3) $\Delta Cmwc-1$ mutant: *CmbrlA* showed delayed induction (reduced at day 4 but upregulated at days 7 and 10). *Cmzf* expression was markedly suppressed at all stages (Fig. S7-B), indicating cross-regulation between *Cmwc-1* and *Cmzf*.

These results (now integrated into Fig. 10 and Supplementary Fig. S7, Page 14-15, Line 254-269) reveal stage-specific disruptions in the conidiation cascade and highlight complex genetic interactions between *CmWc-1*, *CmOzf*, and the core developmental regulators. We believe these data significantly enhance the mechanistic understanding of conidiation control in our study.

4. Minor: - - - - - Importance: BbOzf ... CmOzf Line 138, C2H2 ... C2H2 Line 319, experiments ... studies Line 441, ZnSO4 ... ZnSO4 Line 519, 和 ... ? Line 544-545, Bb ... Cm Line 696: “,” ... “:” Figure 5A, Dolysaccharide ... Polysaccharide

Response: The detailed contents had been revised accordingly throughout the manuscript.

Prof. Jing Han

Editor

Microbiology Spectrum

We are grateful to you and the reviewers for the constructive comments on our manuscript entitled “Light-responsive transcription factor CmOzf integrates conidiation, fruiting body development, and secondary metabolism in *Cordyceps militaris*”. The manuscript has been revised based on the Editorial and Reviewers’ suggestions. Introduction, Results, Discussion, and Materials and Methods have been modified/updated to include the new data as well as in response to specific reviewers’ comments. We feel that the quality of our paper has been substantially improved after revision and appreciate your consideration of this work.

The complete reviewers’ comments and a point-by-point responses are included in the revised submission.

Sincerely yours,

Jinfeng Chen

Reviewer: 3

In this revised version, the authors have addressed some of the previous reviewer comments. However, several important issues remain.

- Line 21: Remove "(C. militaris)".

Response: In line 21, we have removed "(C. militaris)".

- Line 111: Replace "Flug" with "FluG".

Response: Line 111: we have replace "Flug" with "FluG".

- Line 145: Please check the mRNA expression level of the *Cmozf* gene. The reported 2.5-fold increase may not be accurate.

Response: Line 145: The mRNA expression level of *Cmozf* was corrected to 3.5-fold (revised from 2.5-fold).

- Lines 180-181: Please check the fold-change values; they appear unclear or inconsistent.

Response: Lines 180-181 and Lines 184-185, the fold-change values were all revised.

- Figure 6B: Separate the up-regulated and down-regulated genes in the diagram.

Then, perform GO analysis based on this classification. At present, it is difficult to draw clear conclusions from the KEGG or GO analysis.

Response: Lines 197-206: Venn analysis was performed separately for the up-regulated and down-regulated genes, as shown in Figure 6B. Significantly up-regulated gene sets and significantly down-regulated gene sets identified from Figure 6B were subjected to independent Gene Ontology (GO) enrichment analyses, as illustrated in Figure 6C.

- Figure 8: Include the phenotype of the OEC*Cmozf* strain in this figure.

Response: Lines 221-223 and 229-232: The phenotype of the OEC*Cmozf* strain and

its associated gene expression profiles have been incorporated into Figure 8.

- Figure 10: The conclusion that WC-1 represses brlA expression is not clearly supported by the data. Similarly, the claim that Ozf increases abaA expression is also unclear.

- AbaA cannot be simply described as a gene involved in the conidial maturation pathway. Please revise this interpretation.

Response: Lines 254-258: Revised the gene expression experimental results to ensure data consistency and accuracy in the manuscript; removed the uncertain pathways of Ozf-AbaA and WC-1-BrlA based on experimental findings; redesigned the regulatory pathway of Ozf according to the results, as shown in Fig 10-B.

- Figure 2C: Clearly indicate the spores using arrows for better visualization.

Response: White arrows have been added in Figure 2-C to indicate the location of conidia and enhance visual identification.

- Figures (general): Please align text and images uniformly across all figures to improve visual clarity and presentation quality.

Response: The figures and text descriptions have been rechecked and revised to ensure consistency.

Prof. Jing Han

Editor

Microbiology Spectrum

We are grateful to you and the reviewers for the constructive comments on our manuscript entitled "Light-responsive transcription factor CmOzf integrates conidiation, fruiting body development, and secondary metabolism in *Cordyceps militaris*". The manuscript has been revised based on the Editorial and Reviewers' suggestions. Introduction, Results, Discussion, and Materials and Methods have been modified/updated to include the new data as well as in response to specific reviewers' comments. We feel that the quality of our paper has been substantially improved after revision and appreciate your consideration of this work.

The complete reviewers' comments and a point-by-point responses are included in the revised submission.

Sincerely yours,

Jinfeng Chen

Specific comments:

1. The sentence "Light signal significantly affected ... (Fig. 1)." should be followed the subtitle "Cmozf is highly expressed under light exposure."

Response: The legend for Fig.1 has been revised. Please refer to Lines 509-513.

2. The mutants were cultured in the dark on PDA medium for 3 days, followed by 4 days of light exposure (Line 192-194), the authors should explain why the time point (4 days of light exposure) was selected for RNA-seq, and also provide the

time-course expression profiles of *Cmwc-1* in WT (for clarifying the response of *Cmwc-1* to light).

Response: The expression levels of both *Cmwc-1* and *Cmozf* at cultivation time points of 4, 7, and 10 days have been supplemented in Fig. S7. The expression of both genes reached their maximum at day 7 and then began to decrease.

3. For overexpression strains, the authors should make sure which one was used for phenotypic assay (Line 223-226), just like in gene disruption strains.

Response: The screening of the *OECmozf* strain has been added to FIG. S4.

Strain #1 (Δ *Cmwc-1/OECmozf*) and Strain #6 (*OECmozf*) was used for phenotypic assay (Line 220-222).

4. The label of Δ should be unified in all manuscript.

Response: The label of “ Δ ” has been unified in all manuscript.

5. X axis in Figure 6 C and D should be the same (Line 564-569), please revised it.

Response: The X axes in Figure 6 C and D were revised, with the Rich factor range standardized to 0-1.0.

Re: Spectrum01057-25R3 (Light-responsive transcription factor CmOzf integrates conidiation, fruiting body development, and secondary metabolism in *Cordyceps militaris*)

Dear Dr. Jinfeng Chen:

Your manuscript has been accepted, and I am forwarding it to the ASM production staff for publication. Your paper will first be checked to make sure all elements meet the technical requirements. ASM staff will contact you if anything needs to be revised before copyediting and production can begin. Otherwise, you will be notified when your proofs are ready to be viewed.

Sincerely,
Jing Han
Editor
Microbiology Spectrum

Reviewer #4 (Comments for the Author):

accept at current form.